# Progressive Graph Structure Adjustment for Homophily Shift Adaptation

**Hongwei Wen** [1 2]  **Can Zhang** [3]  **Haoyu He** [4]  **Hanyuan Hang** [2]  **Minglong Lei** [3]

## Abstract

We propose *Progressive Structure Adjustment for Homophily Shift* (*PSAHS*), which explicitly addresses cross-domain mismatches in node-level homophily for *Graph Domain Adaptation* (*GDA*). PSAHS increases homophily in the source graph to a prescribed level by reweighting edges and introducing additional intra-class connections for low-homophily nodes, while conservatively refining the target graph based on agreement-consistent predictions from a structure-aware *Graph Neural Network* (*GNN*) and an attribute-only *Multi-Layer Perceptron* (*MLP*) to ensure robustness under label scarcity. After each structural refinement, domain-adversarial training is applied to align node representations across domains. PSAHS adopts a progressive training scheme that alternates between structure adjustment and representation alignment: increasingly informative representations enable safer homophily correction, and the refined structure in turn facilitates improved representation learning. Extensive experiments on multiple GDA benchmarks demonstrate that PSAHS consistently outperforms strong baselines, with particularly pronounced gains under severe homophily mismatch, highlighting the importance of explicit homophily alignment for effective cross-graph transfer.

## 1. Introduction

*Graph Domain Adaptation* (*GDA*) aims to transfer knowledge from a labeled source graph to an unlabeled or sparsely labeled target graph, with applications in cross-network recommendation (Zhao et al., 2025), bioinformatics (Li et al., 2025), and citation analysis (He et al., 2023). A fundamen-

tal challenge in GDA lies in *homophily shift* (Fang et al., 2025b; 2026a), defined as the discrepancy in node label homophily distributions across domains, where node label homophily (Mao et al., 2023) measures the proportion of a node's neighbors sharing the same label. Unlike attribute shift (Fang et al., 2025a) or pairwise label-conditional structure shift (Liu et al., 2024b), node homophily shift directly perturbs the marginal label–structure relationship between node labels and their neighborhoods, which governs message passing in GNNs. As a result, it can severely hinder cross-domain transfer even when node features are well aligned.

Due to sampling bias or platform-specific interaction mechanisms, real-world graphs often exhibit substantially different homophily distributions across domains (Fang et al., 2025b). Moreover, within a single graph, GNN performance varies markedly across nodes with different homophily levels (Ma et al., 2021; Mao et al., 2023), highlighting the critical role of node-level homophily in governing prediction reliability. Nevertheless, most existing GDA methods (Zhu et al., 2020; Fang et al., 2025b) implicitly treat node homophily as a fixed property and address it only indirectly, leaving cross-domain homophily mismatch fundamentally unresolved.

Since node homophily is jointly determined by graph connectivity and node labels, correcting homophily shift requires intervening on one of these two components. Directly altering labels introduces unobservable and irreversible perturbations, and is generally infeasible in the target domain. In contrast, structural adjustment provides a principled and controllable alternative. Indeed, graph structure adjustment has been extensively studied in single-graph settings, including embedding-based structure learning that infers edge strengths from node representations (Yu et al., 2020; Chen et al., 2020; Jiang et al., 2019; Fang et al., 2022), parameterized adjacency learning jointly optimized with GNN parameters (Gao et al., 2019; Franceschi et al., 2019), and graph rewiring strategies based on cluster embeddings or structural-pattern similarities (Li et al., 2023; Suresh et al., 2021). Despite their success in improving in-domain GNN performance, these methods are primarily designed for single-domain scenarios and lack mechanisms to explicitly control node-level homophily. More importantly, under domain shift, embedding or structural similarity does not necessarily imply label similarity, causing such approaches

[1]University of Sydney, Sydney, Australia [2]FUCA AI Labs, Nanjing, China [3]College of Computer Science, Beijing University of Technology, Beijing, China [4]School of Engineering and Applied Science, The George Washington University, Washington, DC, USA. Correspondence to: Minglong Lei <leiml@bjut.edu.cn>.

*Proceedings of the 43rd International Conference on Machine Learning*, Seoul, South Korea. PMLR 306, 2026. Copyright 2026 by the author(s).

to introduce spurious or semantically inconsistent edges. As a result, existing structure learning and reconstruction methods are ill-suited for addressing homophily shift in GDA.

To address these limitations, we treat node homophily not as a passive statistic but as an explicit and controllable structural quantity. Rather than globally rewiring graphs, we selectively adjust only low-homophily nodes in both domains, raising their homophily to a prescribed level while leaving high-homophily nodes unchanged. This targeted intervention eliminates the homophily gap among low-homophily nodes across domains and substantially reduces overall cross-domain homophily mismatch.

In this paper, we propose *Progressive Structure Adjustment for Homophily Shift* (*PSAHS*), which progressively refines graph structures and representations. PSAHS consists of three coordinated modules: (i) *Source-side homophily enhancement*, which suppresses structural noise by reweighting edges and introducing additional intra-class connections for low-homophily nodes using a node-specific adjustment strength with a closed-form solution; (ii) *Target-side homophily alignment*, which addresses homophily mismatch under label scarcity by identifying agreement-consistent nodes via prediction agreement between a structure-aware GNN and an attribute-only MLP, and conservatively refining only low-homophily nodes within this reliable subset; and (iii) *Representation alignment*, which reduces residual cross-domain discrepancy through domain-adversarial training. Modules (ii) and (iii) are applied alternately, forming a progressive training strategy in which informative representations enable safer homophily correction, and the refined structure in turn improves representation learning. PSAHS introduces no additional learnable parameters, scales efficiently to large graphs, and enforces a uniform homophily target across adjusted nodes.

Beyond algorithmic design, a principled theoretical analysis is essential for understanding when and why structural adjustment benefits GDA. Structural interventions directly modify the message-passing operator and may introduce unintended errors if applied improperly. We therefore develop a homophily-aware error analysis that decomposes the target-domain risk into source classification error, cross-domain homophily discrepancy, and representation mismatch, establishing a direct correspondence between each component of PSAHS and a specific error term, and thereby justifying both the necessity and sufficiency of its design.

Our main contributions are as follows. *(i)* We introduce a structural adjustment paradigm that elevates node homophily to prescribed levels in both domains via a node-specific, theory-derived adjustment strength, enabling explicit homophily alignment. *(ii)* We develop a homophily-aware target-domain error analysis that formalizes the ne-

cessity and effectiveness of structure adjustment under homophily shift. *(iii)* Extensive experiments on multiple GDA benchmarks demonstrate that PSAHS consistently outperforms strong baselines, with particularly large gains under severe homophily mismatch.

Our code and implementation details are publicly available at: https://github.com/hongweiaustinwen/PSAHS.

## 2. Preliminaries

### 2.1. Node Classification

A graph is represented as $G = (V, E, X)$, where $V$ is the node set with $|V| = n$, $E$ is the edge set, and $X \in \mathbb{R}^{n \times d}$ is the node attribute matrix, with row $X_u$ denoting the feature vector of node $u$. The graph structure is encoded by an adjacency matrix $A = (A_{uv})_{u,v \in V}$, where $A_{uv} \in \{0, 1\}$ indicates the presence of an edge. We consider node-level classification, where each node $u$ is associated with a label $Y_u \in \mathcal{Y} = [M]$.

Graph neural networks (GNNs) (Wu et al., 2021) learn node representations by aggregating neighborhood information. Given $(X, A)$, an $L$-layer GNN produces node embeddings $Z = \phi(X, A) \in \mathbb{R}^{n \times F}$ via iterative message passing, followed by a classifier $g$ that outputs class probabilities $g(X, A) \in \mathbb{R}^{n \times M}$. For clarity of exposition, we focus on the binary classification case ($M = 2$), noting that all definitions and results extend directly to multi-class settings.

### 2.2. Node Homophily Shift

Graph domain adaptation (GDA) aims to transfer knowledge from a labeled source graph to an unlabeled target graph. Formally, the source domain provides $G_s = (V_s, E_s, X_s)$ with $(X_s, A_s, Y_s) \sim P_s$, while the target domain provides $G_t = (V_t, E_t, X_t)$ with $(X_t, A_t) \sim P_t$. The two domains may differ in node attributes, graph structure, and label distributions.

A key structural property governing GNN behavior is *node (label) homophily*, which measures the tendency of neighboring nodes to share the same label. For a node $u$ with neighborhood $N_u := \{v \in V \mid (u, v) \in E\}$, the homophily ratio is defined as

$$h(u) := \frac{1}{d_u} \sum_{v \in N_u} \mathbf{1}\{Y_u = Y_v\}, \tag{1}$$

where $d_u := |N_u|$ is the node degree. We denote the homophily ratios in the source and target graphs by $h_s(u)$ and $h_t(u)$, respectively, which induce homophily distributions $P_s(h)$ and $P_t(h)$.

**Definition 2.1** (Node Homophily Shift)**.** A *node homophily*

*shift* occurs when the source and target homophily distributions differ, i.e., $P_s(h) \neq P_t(h)$.

Such homophily shifts have been widely observed in real-world graphs and are known to critically affect message-passing GNNs under domain shift (Fang et al., 2025b).

## 2.3. Graph Structure Adjustment

Real-world graphs often contain noisy or task-irrelevant edges, motivating explicit graph structure adjustment. Given a graph $G = (V, E, X)$ with adjacency matrix $A$, we define an adjusted graph by replacing $A$ with an alternative matrix $A' = (A'_{uv})_{u,v \in V}$ of the same dimension. The adjusted adjacency may reweight existing edges, remove uninformative ones, or introduce new connections, and edge weights are not restricted to be binary.

Under the adjusted structure, node representations are computed as $Z' = \phi(X, A')$, and predictions are given by $g(X, A')$. When labels $Y$ are available, the node homophily under $A'$ is defined as

$$h'(u) := \frac{\sum_{v \in N'_u} A'_{uv} \mathbf{1}\{Y_v = Y_u\}}{\sum_{v \in N'_u} A'_{uv}}, \tag{2}$$

where $N'_u$ is the adjusted neighborhood of node $u$ induced by $A'$. In general, this adjusted homophily differs from the original homophily $h(u)$ under the observed graph.

## 3. Progressive Structure Adjustment for Homophily Shift (PSAHS)

GNNs propagate information through neighborhood aggregation. For low-homophily nodes—i.e., nodes connected to many neighbors of different classes—message passing mixes incompatible signals and pushes representations away from class-consistent regions. This behavior is not an architectural artifact but a structural effect of neighborhood composition. Under domain shift, the proportions of high- and low-homophily nodes may differ substantially across domains, giving rise to node homophily shift. Most existing GDA methods align node features or learned representations while implicitly assuming reliable graph structures. However, under homophily shift, mismatched neighborhood composition biases the message-passing operator itself, which cannot be corrected by feature-level alignment alone. This observation motivates explicit intervention at the graph-structure level.

Our central principle therefore follows naturally: before enforcing representation invariance across domains, graph structures should be explicitly corrected to ensure that low-homophily nodes are sufficiently connected to same-class neighborhoods. Based on this principle, we propose *PSAHS*, which consists of three components: (i) *source-side ho-*

*mophily enhancement*, (ii) *target-side homophily alignment*, and (iii) *cross-domain representation alignment*. The latter two interact in a progressive loop, where structural correction stabilizes message passing and enables increasingly accurate homophily estimation and safer updates.

### 3.1. Source-side Structure Adjustment for Homophily Enhancement

We first perform a one-time structural refinement on the source graph. Since ground-truth labels are available, aggregation bias induced by heterophilous or label-inconsistent neighborhoods can be corrected deterministically, yielding a stable homophily reference for cross-domain adaptation.

Let $h \in (0, 1]$ denote a prescribed node homophily level. For a source node $u$, if $h_s(u) \geq h$, its incident edges remain unchanged; otherwise, its local structure is adjusted to raise its homophily to exactly $h$. This selective normalization avoids unnecessary intervention while aligning all low-homophily nodes to a common reference.

Let $A'_s$ denote the refined adjacency matrix. For nodes with $h_s(u) < h$, we upweight homophilous edges, downweight heterophilous edges, and add new homophilous edges incident to $u$:

$$(A'_s)_{uv} := \begin{cases} 1 + \alpha_u, & (A_s)_{uv} = 1, \ (Y_s)_u = (Y_s)_v, \\ 1 - \alpha_u, & (A_s)_{uv} = 1, \ (Y_s)_u \neq (Y_s)_v, \\ \alpha_u, & (A_s)_{uv} = 0, \ v \in \tilde{N}_u, \\ (A_s)_{uv}, & \text{otherwise.} \end{cases} \tag{3}$$

Here, $\alpha_u$ is a node-specific adjustment strength. The augmented neighborhood $\tilde{N}_u$ contains $|\tilde{N}_u| = d_u(1 - h_s(u))$ same-class nodes that are uniformly randomly sampled from the non-neighbors of $u$, matching the number of heterophilous neighbors and ensuring balanced compensation.

Let $d_u$ denote the degree of $u$. After refinement, the effective homophilic and heterophilic contributions are

$$S'_u := \sum_{v \in N'_u} (A'_s)_{uv} \mathbf{1}\{Y_v = Y_u\}$$
$$= (1 + \alpha_u) d_u h_s(u) + \alpha_u d_u (1 - h_s(u)),$$
$$D'_u := \sum_{v \in N'_u} (A'_s)_{uv} \mathbf{1}\{Y_v \neq Y_u\} = (1 - \alpha_u) d_u (1 - h_s(u)).$$

The adjusted homophily defined in (2) is given by

$$h'_s(u) = \frac{S'_u}{S'_u + D'_u} = \frac{\alpha_u + h_s(u)}{1 + \alpha_u h_s(u)}. \tag{4}$$

Enforcing $h'_s(u) = h$ yields the unique closed-form solution

$$\alpha_u = \frac{h - h_s(u)}{1 - h \cdot h_s(u)}. \tag{5}$$

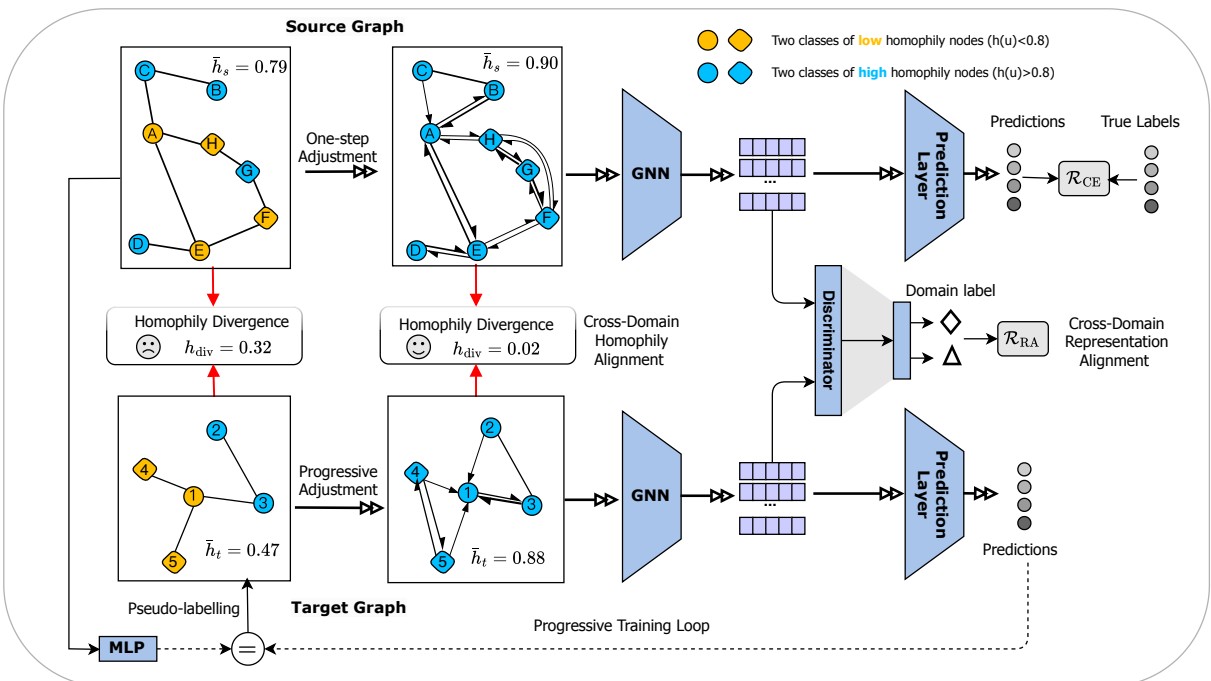

*Figure 1.* Framework of Progressive Structure Adjustment for Homophily Shift (PSAHS). The top and bottom flows illustrate the source and target domains, respectively. In the source domain, low-homophily nodes are adjusted by raising them to a prescribed level $h = 0.8$, resulting in a directed graph that is more homophilous ($\bar{h}_s = \frac{1}{n_s}\sum_u h_s(u) : 0.79 \rightarrow 0.90$). In the target domain, the MLP–GNN agreement mechanism is applied to refine node representations, reducing cross-domain homophily divergence from $h_{\mathrm{div}} = |\bar{h}_t - \bar{h}_s| : 0.32 \rightarrow 0.02$. Finally, the aggregated representations from the adjusted graphs are aligned via adversarial alignment, and updated target-domain predictions are iteratively fed back to further enhance the structural refinement.

If two connected nodes $u$ and $v$ have different homophily ratios, their adjustment strengths differ, i.e., $\alpha_u \neq \alpha_v$. According to (3), this asymmetry leads to $(A'_s)_{uv} \neq (A'_s)_{vu}$, implying that the adjusted graph becomes directed. Based on the directed graph $A'_s$, we adopt a row-normalized message passing to generate the $\ell$-th layer representation,

$$Z_u^{(\ell)} = \sigma\Big(D_{uu}^{-1}\sum_v (A'_s + I)_{uv} Z_v^{(\ell-1)} W^{(\ell)}\Big),$$

where $\sigma(\cdot)$ is the activation function, $D_{uu} = \sum_v (A'_s + I)_{uv}$ and $W^{(\ell)}$ is the $\ell$-th layer weight matrix in GNNs.

### 3.2. Target-side Structure Adjustment for Homophily Alignment

In the target domain, homophily cannot be directly computed due to missing labels. Applying (3) with unreliable pseudo-labels may inject structural errors, particularly for nodes with sparse or heterogeneous neighborhoods. The key challenge is therefore to determine where and when structural adjustment can be safely applied.

PSAHS addresses this challenge by restricting target-side refinement to nodes satisfying cross-model agreement—i.e., whose predicted labels are consistent between a source-

pretrained MLP and the continuously updated GNN.

$$V_t^{\mathrm{a}} = \{u \in V_t \mid \widehat{Y}_u^{\mathrm{GNN}} = \widehat{Y}_u^{\mathrm{MLP}}\}. \qquad (6)$$

For $u \in V_t^{\mathrm{a}}$, the pseudo-label is $\widehat{Y}_u = \widehat{Y}_u^{\mathrm{GNN}} = \widehat{Y}_u^{\mathrm{MLP}}$. Homophily is estimated using only agreement neighbors:

$$\widehat{h}_t(u) = \frac{1}{d_u^{\mathrm{a}}} \sum_{v \in N_u \cap V_t^{\mathrm{a}}} \mathbf{1}\{\widehat{Y}_v = \widehat{Y}_u\}, \qquad (7)$$

where $d_u^{\mathrm{a}} = |N_u \cap V_t^{\mathrm{a}}|$. This estimator suppresses noisy or isolated predictions and reflects mutually consistent neighborhood evidence.

Nodes in $V_t^{\mathrm{a}}$ with $\widehat{h}_t(u) < h$ are refined using (3), where the adjustment strength $\alpha_u$ is computed according to (5) with $h_s(u)$ replaced by $\widehat{h}_t(u)$, and the augmented neighbors $\widetilde{N}_u$ is formed by $d_u(1 - \widehat{h}_t(u))$ same-class, originally non-neighbor nodes in $V_t^{\mathrm{a}}$ with the highest homophily, while nodes outside $V_t^{\mathrm{a}}$ remain unchanged. As pseudo-label reliability improves, we perform more stable graph adjustments, enabling effective alignment of target-domain homophily with the source domain.

### 3.3. Cross-domain Representation Alignment

Structural alignment stabilizes message passing but does not eliminate all cross-domain discrepancies arising from

feature statistics or higher-order effects. PSAHS therefore incorporates domain-adversarial training to align residual representation mismatch.

We jointly learn a GNN encoder $\phi$ and classifier $g$ by

$$\min_{\phi,g} \ \mathcal{R}_{\mathrm{CE}}(g) + \gamma_{\mathrm{RA}}\mathcal{R}_{\mathrm{RA}}(\phi), \tag{8}$$

where $\mathcal{R}_{\mathrm{CE}}(g) = -\frac{1}{n_s} \sum_{u \in V_s} \mathcal{L}_{\mathrm{CE}}\big(g_u(X_s, A'_s), Y_u\big)$, and

$$\mathcal{R}_{\mathrm{RA}}(\phi) = \max_{\xi:\mathbb{R}^F \to (0,1)} \Big[\frac{1}{n_s} \sum_{u \in V_s} \log \xi(\phi_u(X_s, A'_s))$$
$$+ \frac{1}{n_t} \sum_{u \in V_t} \log(1 - \xi(\phi_u(X_t, A'_t)))\Big]. \tag{9}$$

Here, $\xi$ denotes a domain discriminator and $\mathcal{L}_{\mathrm{CE}}$ is the cross-entropy loss. Adversarial training encourages representations to be domain-invariant while remaining discriminative for the source task.

### 3.4. Progressive Training Procedure

Source-side adjustment is performed once using ground-truth labels, yielding a stable reference structure. Target-side adjustment relies on pseudo-labels whose reliability evolves over training. Performing structure refinement in a single step is therefore ill-posed, as early pseudo-label errors may lead to incorrect structural changes.

To address this issue, PSAHS adopts a progressive training strategy that alternates between structure adjustment and representation alignment. Instead of performing graph refinement in a single step, PSAHS incrementally updates the target structure as representation quality improves. At each iteration, GNN representations learned under the current graph structure become increasingly informative, enabling more accurate pseudo-labels and more reliable homophily estimation. These improved estimates guide safer and more effective structure refinement for aligning cross-domain homophily. In turn, the refined graph structure reduces aggregation noise and cross-domain bias in message passing, allowing the GNN model to learn more discriminative representations in subsequent iterations. Through this iterative and progressive process, PSAHS stabilizes target-side graph refinement and continuously enhances adaptation performance.

## 4. Theoretical Analysis

In this section, we develop a theoretical framework to analyze the generalization behavior of PSAHS and to identify which components of the target error are explicitly reduced by controlling node-level homophily. Our analysis establishes a one-to-one correspondence between the three modules of PSAHS and the three error terms in a target risk

bound, providing principled justification for the algorithm design. To isolate structural effects from architectural complexity, we follow Mao et al. (2023) and adopt the *Simplified Graph Network* (*SGN*) model (Wu et al., 2019), where node representations are computed as $Z' = (D')^{-1}A'X$, followed by an MLP classifier, and $D'$ is the degree matrix of $A'$. This abstraction allows us to focus on how graph structure influences generalization. Under SGN, structure adjustment affects learning through two mechanisms: *(i)* modifying aggregated representations $Z'$ via neighborhood composition and edge weights, and *(ii)* reshaping node homophily ratios $h'(u)$, which determine the proportion of same-class signals contributing to aggregation. These two effects underpin the following error decomposition.

### 4.1. Target Error Decomposition under Adjusted Structures

We first characterize the target-domain classification error under adjusted graph structures.

**Theorem 4.1** (Target Risk Bound under SGN). *Let Assumptions A.1, A.2, A.3 in Appendix A.1 hold. Under SGN, for any adjusted graph structures $(A'_s, A'_t)$, any $\delta \in (0,1)$ and $\gamma > 0$, with probability at least $1 - \delta$, the target risk $\mathcal{R}_t(g)$ satisfies*

$$\mathcal{R}_t(g) \le \widehat{\mathcal{R}}_s^\gamma(g) + W_1(h'_s, h'_t) + \mathrm{IPM}(Z'_s, Z'_t)$$
$$+ \sqrt{\ln(1/\delta)/n_s} + \sqrt{\ln(1/\delta)/n_t}. \tag{10}$$

*Here, $\widehat{\mathcal{R}}_s^\gamma(g)$ is the empirical source margin loss, $W_1(h'_s, h'_t)$ is the 1-Wasserstein distance between node homophily distributions,*

$$W_1(h'_s, h'_t) := \frac{1}{n_s n_t} \sum_{u \in V_s} \sum_{v \in V_t} |h'_s(u) - h'_t(v)|, \tag{11}$$

*and $\mathrm{IPM}(Z'_s, Z'_t)$ measures representation discrepancy,*

$$\mathrm{IPM}(Z'_s, Z'_t) := \frac{1}{n_s n_t} \sum_{u \in V_s} \sum_{v \in V_t} \|(Z'_s)_u - (Z'_t)_v\|_2. \tag{12}$$

Detailed assumptions and proofs are deferred to Appendix A.1. The bound in (10) decomposes the target error into three terms, which we analyze next.

### 4.2. Source-Domain Error Reduction

We first analyze the effect of source-side structure adjustment on the empirical source margin loss $\widehat{\mathcal{R}}_s^\gamma(g)$.

**Theorem 4.2.** *For $a, b \in \mathbb{R}$, let $a \vee b := \max\{a, b\}$. Let $g$ be the SGN classifier trained on the adjusted source graph. Then the empirical source margin loss satisfies*

$$\widehat{\mathcal{R}}_s^\gamma(g) = 2 - \frac{2}{n_s} \sum_{u \in V_s} (h_s(u) \vee h), \tag{13}$$

*which decreases monotonically with the homophily level $h$.*

Setting $h = 0$ corresponds to applying no source-domain graph adjustment. Consequently, any $h > 0$ strictly reduces $\widehat{\mathcal{R}}_s^\gamma(g)$, indicating that the proposed graph adjustment tightens the first term in (10).

### 4.3. Cross-domain Homophily Discrepancy Reduction

We now analyze how target-side structural adjustment reduces the cross-domain homophily discrepancy $W_1(h_s', h_t')$, starting from the fully supervised setting where target-domain labels are available.

**Theorem 4.3.** *For $a, b \in \mathbb{R}$, let $a \wedge b := \min\{a, b\}$. Assume that target-domain labels are known. If both the source and target graphs are adjusted according to (3) with a shared threshold $h \in (0, 1]$ and node-specific strengths $\alpha_u$ defined in (5), then $W_1(h_s', h_t') = W_1(h_s, h_t) - W_1(h \wedge h_s, h \wedge h_t)$.*

This identity directly yields $W_1(h_s', h_t') \leq W_1(h_s, h_t)$, with strict inequality whenever $h$ exceeds the minimum node-wise homophily, showing that the proposed structural adjustment strictly contracts the cross-domain homophily discrepancy in the fully supervised setting.

In practice, target labels are unavailable. To avoid the structural errors caused by the false pseudo-labels, PSAHS restricts adjustment to the agreement set $V_t^a$. Let

$$E_G(u) := \{\widehat{Y}_u^{\mathrm{GNN}} \neq Y_u\}, \quad E_M(u) := \{\widehat{Y}_u^{\mathrm{MLP}} \neq Y_u\}$$

denote the prediction error events of the GNN and MLP on a target node $u$, with marginal error rates

$$\varepsilon_G := P(E_G(u)), \quad \varepsilon_M := P(E_M(u)).$$

**Theorem 4.4** (Noise filtering via prediction agreement). *Let the agreement set $V_t^a$ be as in (6) and $p_A := P(u \in V_t^a) > 0$. Moreover, assume*

$$P(E_G(u) \cap E_M(u)) \leq \rho\, \varepsilon_G \varepsilon_M, \qquad \rho \geq 1. \quad (14)$$

*Then the pseudo-label error rate on $V_t^a$ satisfies*

$$P(\widehat{Y}_u \neq Y_u \mid u \in V_t^a) \leq \frac{\rho\, \varepsilon_G \varepsilon_M}{p_A}. \quad (15)$$

Assumption (14) is mild here, since GNN errors are mainly induced by structure-dependent aggregation bias, while MLP errors arise from attribute-level covariate shift, making their joint failure a second-order event.

For $v \in \mathcal{N}_u$, define the agreement separation gap

$$\varepsilon_s = \left| P(v \in V_t^a \mid Y_v = Y_u) - P(v \in V_t^a \mid Y_v \neq Y_u) \right|,$$

which quantifies how strongly the agreement set favors same-class neighbors over different-class neighbors.

**Theorem 4.5** (Homophily Estimation Error). *Fix $u \in V_t^a$ with $d_u^a > 0$ and $p_A := P(u \in V_t^a) > 0$. Then for any $\delta \in (0, 1)$, with probability at least $1 - \delta$,*

$$|\widehat{h}_t(u) - h_t(u)| \leq \frac{2\rho\, \varepsilon_G \varepsilon_M}{p_A} + \sqrt{\frac{2\ln(2/\delta)}{d_u^a}} + \frac{h_t(u)}{p_A}\varepsilon_s.$$

For low-homophily nodes, $\varepsilon_s$ is typically small since heterogeneous neighborhood mixing contaminates message passing, rendering prediction agreement nearly class-agnostic. As training proceeds, $\varepsilon_G, \varepsilon_M$ decrease and $d_u^a$ grows, yielding increasingly accurate homophily estimates.

Combining the analysis in Theorems 4.3 and 4.5, we derive the following upper bound of the cross-domain homophily discrepancy after the agreement-based structure refinement in the target domain.

**Theorem 4.6.** *Let $h_t'$ denote the homophily under pseudo-label-based refinement. Assume that every target node $u$ satisfies $d_u^a > 0$ and $p_A := P(u \in V_t^a) > 0$. Then for any $\delta \in (0, 1)$, with probability at least $1 - \delta$,*

$$W_1(h_s', h_t') \leq W_1(h_s, h_t) - W_1(h \wedge h_s, h \wedge \widehat{h}_t)$$

$$+ \frac{1}{n_t} \sum_{u \in V_t} \left( \frac{4\rho\, \varepsilon_G \varepsilon_M}{p_A} + \sqrt{\frac{8\ln(\frac{2}{\delta})}{d_u^a}} + \frac{h_t(u) + h_t'(u)}{p_A}\varepsilon_s \right).$$

This ensures a reduction of the homophily discrepancy $W_1(h_s', h_t')$ under small pseudo-label error and low agreement separation gap.

### 4.4. Representation Discrepancy Reduction via IPM Alignment

We finally analyze how cross-domain representation alignment in PSAHS reduces the representation discrepancy term in the target risk bound (10) via adversarial alignment.

**Theorem 4.7.** *Let $Z_s' = \phi(X_s, A_s')$ and $Z_t' = \phi(X_t, A_t')$ denote node representations aggregated over the adjusted source and target graphs, respectively. Let Assumptions A.5 and A.6 in Appendix A.4 hold. Then minimizing the adversarial objective (8) decreases the representation discrepancy $\mathrm{IPM}(Z_s', Z_t')$ defined in (12).*

Theorem 4.7 shows that adversarial alignment contracts the cross-domain representation mismatch $\mathrm{IPM}(Z_s', Z_t')$, thereby complementing the mitigation of structure-induced mismatch achieved by homophily-aware graph adjustment.

### 4.5. Complexity Analysis

PSAHS is computationally efficient, as it avoids dense structure learning and introduces no additional learnable parameters. Source-side adjustment is performed once with cost $O(n_s + |E_s|)$. Target-side structure refinement is applied periodically and incurs

$$O\big(LF^2 n_t + LF|E_t| + (\theta_t n_t)\log(\theta_t n_t)\big)$$

per update, where $\theta_t \in (0, 1]$ is the agreement-set ratio. All structural updates are node-wise, closed-form, and restricted

to a subset of nodes, resulting in minimal overhead beyond standard GNN training. Further details of the complexity analysis and empirical scalability results are provided in Appendix A.5.

# 5. Experiments

**Baselines.** We compare our approach PSAHS against the following representative baselines: feature alignment methods UDA-GCN (Wu et al., 2020), ASN (Zhang et al., 2021), GraphAlign (Huang et al., 2024), ADAlign (Chen et al., 2026) and JHGDA (Shi et al., 2023); structure-shift methods StruRW (Liu et al., 2023) and PairAlign (Liu et al., 2024b); and the homophily-based method HGDA (Fang et al., 2025b).

**Experimental Settings.** For evaluation, the source graph is used for training, 20% of target node labels for validation, and the remaining 80% for testing. Hyperparameters are selected based on target validation performance, and accuracy on target test nodes is reported. Additional hyperparameter details are provided in Appendix B.2.

**Synthetic Experiments.** We evaluate PSAHS under varying node homophily shifts using SBM-simulated graphs with three balanced classes. Node features are sampled from class-specific 10-dimensional Gaussians, with shifted means between source and target domains and randomly rotated diagonal covariance matrices. To induce homophily shift, one domain fixes intra-/inter-class edge probabilities ($p = 0.02$, $q = 0.002$, homophily 0.832), while the other progressively reduces homophily by randomly removing homophilous edges and adding heterogeneous ones until the homophily decreases from 0.8 to 0.1. Detailed data generation procedures and visualizations of the attribute distributions are provided in Appendix B.1.1.

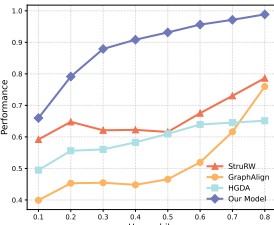 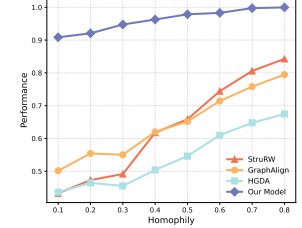

(a) Varied target homophily    (b) Varied source homophily

*Figure 2.* Accuracy under different homophily settings.

Figure 2 reports the GDA accuracy on synthetic datasets. PSAHS consistently outperforms all baselines under varying node homophily shifts, regardless of whether the source or target domain has higher homophily, demonstrating its effectiveness in mitigating homophily shift and supporting the theoretical analysis in Section 4. Performance differences mainly arise from how methods handle homophily shift. Methods without explicit homophily correction (e.g., GraphAlign and StruRW) degrade rapidly as homophily diverges. HGDA shows improved stability through attribute–

topology alignment but fails under severe shifts due to the lack of direct homophily revision. In contrast, PSAHS maintains robust performance by directly correcting graph structure for low-homophily nodes, addressing homophily mismatch at its structural source.

**Real-world Experiments.** We evaluate PSAHS on four real-world benchmarks: `Citation`, `Airport`, `Blog`, and `Twitch`. `Citation` includes two citation networks, `DBLPv8` (D) and `ACMv9` (A); `Airport` contains air-traffic networks from `USA` (U), `Brazil` (B), and `Europe` (E); `Twitch` consists of six regional gamer networks (`DE`, `EN`, `ES`, `FR`, `PT`, `RU`); and `Blog` contains two disjoint social graphs, `Blog1` and `Blog2`. Dataset statistics are provided in Appendix B.1.2.

Tables 1 and 2 show that PSAHS consistently outperforms all baselines on 15 GDA tasks, achieving improvements of up to 21.94% on `B2-B1`. These results highlight the benefit of jointly enhancing source homophily and mitigating cross-domain homophily shift, especially on low-homophily datasets such as `Blog` (average homophily 0.38). In contrast, methods operating on the original graphs perform poorly, as low homophily degrades message passing and homophily mismatch further hinders cross-domain transfer.

**Reliable Pseudo-labeling via Model Agreement.** Since true target labels are unavailable, pseudo-labels are crucial for target graph refinement. PSAHS uses agreement-consistent predictions from a GNN and an MLP as pseudo-labels and refines edges only for nodes with such agreement. We compare four alternatives: GNN_PL (all GNN pseudo-labels), MLP_PL (all MLP pseudo-labels), Curriculum_PL (progressively refining from the top 20% to 80% most confident nodes), and Prototype_PL (prototype-based denoising with online-updated class centers).

Figure 3 shows that the proposed GNN–MLP agreement strategy consistently achieves the best performance on `Blog`. Using only GNN pseudo-labels is sensitive to structural bias, while MLP-based labels lack structural awareness. Curriculum- and prototype-based methods partially reduce noise but still depend on single-model predictions. In contrast, PSAHS restricts refinement to nodes where structure-aware GNN and attribute-only MLP agree, yielding more reliable pseudo-labels and stable, superior performance under varying homophily shifts.

**Analysis of Prescribed Homophily Level $h$.** We study the sensitivity of PSAHS to the prescribed homophily level $h$. As shown in Figure 4, PSAHS exhibits smooth and regular performance trends as $h$ varies, with monotonic or mildly unimodal behavior and no abrupt oscillations or performance collapse. Across all tasks, the curves remain flat in a wide neighborhood around their optima, indicating that PSAHS is insensitive to the exact choice of $h$ and admits a broad, stable optimal region rather than a narrow peak. As

*Table 1.* Performance on DBLP/ACM and Airport datasets.

| Models | Citation | | Airport | | | | | |
|---|---|---|---|---|---|---|---|---|
| | A-D | D-A | U-E | E-U | B-E | E-B | B-U | U-B |
| UDAGCN | 68.86±1.08 | 63.91±1.16 | 48.87±1.45 | 43.41±0.52 | 50.77±0.70 | 47.62±1.13 | 49.78±0.55 | 61.22±1.25 |
| ASN | 72.70±0.38 | 71.62±0.78 | 46.45±0.34 | 46.25±2.65 | 49.62±1.04 | 59.03±1.56 | 49.86±2.37 | 51.91±0.89 |
| JHGDA | 75.58±1.65 | 73.22±0.68 | 50.75±3.01 | 52.27±6.25 | 56.64±1.14 | 73.13±1.37 | 50.20±0.54 | 69.27±3.32 |
| StruRW | 70.19±2.10 | 66.57±0.42 | 53.77±0.98 | 49.67±2.88 | 56.06±1.31 | 65.65±0.15 | 52.19±2.01 | 62.84±0.11 |
| PairAlign | 75.24±1.06 | 74.77±1.57 | 55.39±0.94 | 54.28±2.07 | 55.72±0.97 | 52.90±1.48 | 52.78±1.62 | 67.86±0.50 |
| GraphAlign | 78.65±1.07 | 75.06±1.61 | 54.32±0.86 | 57.34±1.39 | 58.80±0.92 | 73.12±0.89 | 54.38±1.11 | 62.90±1.63 |
| HGDA | 79.10±0.81 | 75.60±0.89 | 57.20±1.12 | 57.00±1.04 | 58.40±0.71 | 72.10±0.92 | 56.90±2.01 | 72.10±4.80 |
| ADAlign | 78.49±7.70 | 73.47±1.08 | 48.62±2.17 | 49.38±0.33 | 53.08±0.89 | 60.15±0.57 | 50.81±0.91 | 70.99±5.29 |
| **PSAHS** | **82.61±0.39** | **76.32±0.44** | **59.20±0.47** | **58.43±0.76** | **60.03±0.49** | **74.34±0.34** | **58.06±0.72** | **73.08±0.66** |

*Table 2.* Performance on Blog and Twitch datasets.

| Models | Blog | | Twitch | | | | |
|---|---|---|---|---|---|---|---|
| | B1-B2 | B2-B1 | DE-EN | DE-ES | DE-FR | DE-PT | DE-RU |
| UDAGCN | 47.10±1.23 | 46.80±1.16 | 53.97±1.34 | 57.49±1.21 | 54.53±1.42 | 55.32±1.13 | 63.59±1.50 |
| ASN | 63.20±1.08 | 52.40±1.62 | 52.58±1.29 | 54.68±1.47 | 52.79±1.31 | 56.03±1.20 | 66.18±1.44 |
| JHGDA | 61.90±1.12 | 64.30±1.99 | 55.80±0.98 | 62.35±2.01 | 59.21±1.46 | 62.85±1.25 | 72.05±0.93 |
| StruRW | 63.59±1.09 | 62.64±0.96 | 54.81±1.02 | 66.03±1.52 | 60.48±1.67 | 63.96±1.14 | 72.27±1.18 |
| PairAlign | 66.20±0.78 | 65.40±1.29 | 56.69±0.59 | 65.29±2.03 | 57.52±1.65 | 62.50±1.03 | 73.28±0.82 |
| GraphAlign | 47.14±0.99 | 45.83±1.04 | 56.02±0.71 | 69.04±0.74 | 62.46±1.10 | 65.74±0.57 | 71.79±0.81 |
| HGDA | 68.30±0.76 | 67.70±1.28 | 49.93±1.60 | 54.43±0.94 | 54.94±1.09 | 48.25±1.62 | 54.60±1.11 |
| ADAlign | 64.69±0.23 | 64.90±0.32 | **60.21±0.23** | 59.95±1.52 | 53.25±3.55 | 57.73±1.23 | 50.14±1.64 |
| **PSAHS** | **88.05±0.52** | **89.64±0.49** | 57.97±0.46 | **71.29±0.65** | **64.63±0.36** | **66.84±0.62** | **74.13±0.59** |

The best and second-best performances are marked as **bold** and underline, respectively.

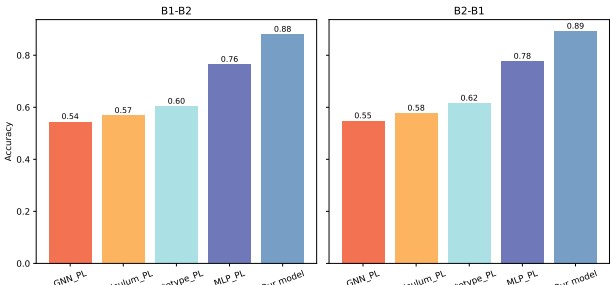

*Figure 3.* GDA performance of different PL strategies.

a result, $h$ can be reliably tuned using a standard validation set without delicate hyperparameter search.

We further observe that the optimal $h$ is not always maximal. While performance on datasets such as Blog improves steadily with larger $h$, datasets such as Airport favor an intermediate value $h < 1$. This reflects a trade-off induced by pseudo-label noise: overly aggressive homophily enforcement may introduce spurious edges when pseudo-labels are imperfect. Importantly, this effect leads to gradual rather than abrupt degradation, highlighting the robustness of PSAHS with respect to $h$. A detailed analysis of this trade-off is provided in Theorem B.2 (Appendix B.6).

**Analysis of Agreement Set.** We further analyze the behavior of the agreement set used for structure refinement. Rather than being a sensitive hyperparameter, the agreement set acts as an adaptive confidence mechanism whose size is determined by transfer difficulty. Empirically, easier transfer scenarios naturally produce larger agreement sets, while more challenging shifts lead to smaller but more reliable

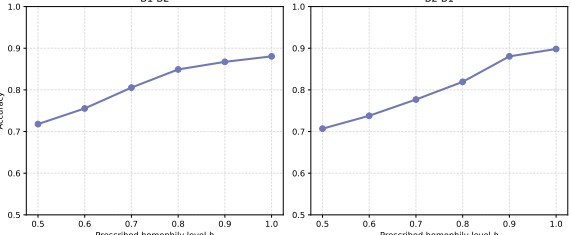

(a) Performance on Blog with varied $h$.

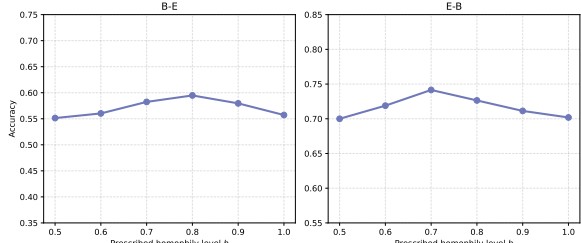

(b) Performance on B and E of Airport with varied $h$.

*Figure 4.* Parameter analysis of prescribed homophily level $h$.

subsets. As shown in Figure 5, the agreement ratio (defined as the proportion of agreement nodes in the target graph) evolves automatically during training, increasing from approximately $17\% - 76\%$ in early epochs to $68\% - 83\%$ in later stages. This indicates that improved representations gradually increase prediction consistency and expand the set of confident target nodes.

Importantly, agreement filtering produces highly reliable pseudo-labels by retaining only nodes on which the source-trained MLP and the current GNN predict the same class. The resulting agreement set achieves consistently high ac-

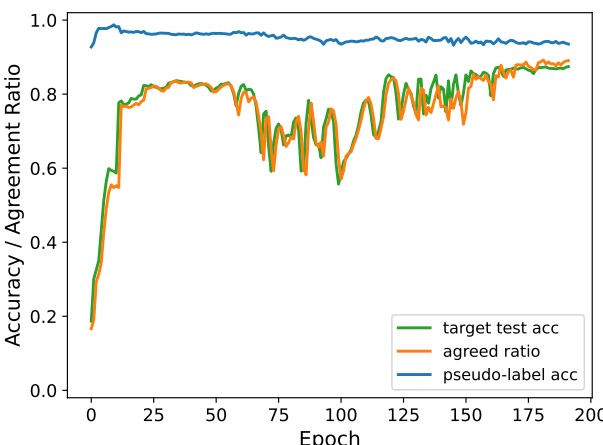

*Figure 5.* Agreement Set Analysis.

curacy, reaching approximately $98.5\%$ in early training and remaining around $95\% - 97\%$ afterwards. This observation is supported by Theorem 4.4, which shows that the pseudo-label error on the agreement set reduces to a second-order term, $\varepsilon_G \varepsilon_M$, yielding substantially lower error than standard pseudo-labeling. Consequently, the agreement mechanism naturally realizes a precision–coverage trade-off: early training emphasizes high precision with limited coverage to avoid unreliable structural updates, while later stages gradually expand coverage while maintaining strong reliability, enabling effective large-scale refinement. Empirically, PSAHS remains stable throughout this process and continuously benefits from increasing agreement coverage. Importantly, these gains are achieved through controlled and distributed structural refinement rather than aggressive graph rewiring; detailed statistics of graph modification are provided in Appendix B.4.

**Ablation Studies.** We study three PSAHS variants to assess the impact of structure adjustment across domains: *w/o adjustment* (no structural refinement), *w/o source* (adjusting only the target graph), and *w/o target* (adjusting only the source graph). Table 3 shows that both single-graph variants outperform *w/o adjustment*, indicating that adjusting either domain alone benefits GDA. Notably, the full PSAHS model achieves the best performance, demonstrating the advantage of jointly enhancing homophily and mitigating cross-domain node homophily shift.

*Table 3.* Ablation study on `Blog` and `Airport` datasets.

| Models | Blog | | Airport | | | | | |
|---|---|---|---|---|---|---|---|---|
| | B1-B2 | B2-B1 | U-E | E-U | B-E | E-B | B-U | U-B |
| w/o adjustment | 54.30 | 56.25 | 49.33 | 47.76 | 50.99 | 67.54 | 50.62 | 65.47 |
| w/o source | 82.10 | 82.88 | 55.87 | 54.66 | 55.58 | 69.86 | 52.56 | 70.75 |
| w/o target | 61.66 | 60.17 | 52.42 | 56.07 | 54.34 | 72.75 | 54.08 | 69.18 |
| **PSAHS** | **88.05** | **89.64** | **59.20** | **58.43** | **60.03** | **74.34** | **58.06** | **73.08** |

# 6. Conclusion

In this paper, we investigated the challenge of *node homophily shift* in GDA, a structural mismatch that hinders cross-domain transfer even when feature distributions are well aligned. To address this issue, we proposed a progressive structure adjustment framework that performs a one-time source-side homophily enhancement using ground-truth labels, and then iteratively alternates between target-side homophily alignment guided by reliable pseudo-labels and cross-domain representation alignment via adversarial training. Our theoretical analysis established an explicit connection between node-level homophily distributions and the target-domain error bound, thereby motivating structural refinement as a principled and controllable approach to domain adaptation. Extensive experiments on both synthetic and real-world benchmarks demonstrated that the proposed method consistently outperforms strong baselines, with particularly large improvements under severe homophily mismatch. These results highlight the critical role of explicit node-level structural alignment in enabling effective cross-graph transfer. In the future, we will explore extending our agreement-based graph refinement framework to multi-view graph domain adaptation (Fang et al., 2026b) and open-set graph domain adaptation (Shen et al., 2025).

# Impact Statement

This work studies graph domain adaptation under node homophily shift, with the primary goal of improving the generalization and reliability of graph neural networks when transferring across graph-structured datasets. The proposed method is evaluated on standard benchmarks in citation, social, and infrastructure networks and is intended for general machine learning research and data analysis tasks. The approach does not introduce new data sources, does not infer sensitive attributes, and does not target high-stakes decision-making scenarios.

Potential risks mainly arise from incorrect pseudo-labels in the target domain, which may affect structural adjustment and propagate errors during adaptation. To mitigate this risk, the method adopts a conservative agreement-based strategy and performs structure refinement progressively.

From a theoretical perspective, our analysis is conducted under the Simplified Graph Network (SGN) abstraction rather than general multi-layer GNNs. While this limits direct characterization of architectural effects, SGN intentionally isolates the structural propagation mechanism and corresponds to the linearized message-passing core of GNNs. This enables us to attribute generalization behavior directly to graph structure and homophily, providing theoretical insight into graph domain adaptation. Extending the analysis to more expressive architectures remains future work.

Overall, this work is expected to have a positive impact by improving the stability and interpretability of graph-based learning under distribution shift, while posing minimal risk beyond those commonly associated with graph representation learning.

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

This appendix complements the main text by providing detailed theoretical derivations, additional proofs, and extended experimental results that support the main claims of the paper. Specifically, Appendix A collects all formal proofs omitted from the main text for clarity. Appendix A.1 presents the complete proof of the target-domain error bound in Theorem 4.1, including all intermediate steps, assumptions, and constants. Appendix A.2 provides a detailed analysis of how source-side structure adjustment reduces the empirical source margin loss. Appendix A.3 contains proofs related to homophily discrepancy reduction, including the contraction property under ideal adjustment, the effect of agreement-based pseudo-labeling, and the stability of homophily alignment. Appendix A.4 rigorously justifies that adversarial representation alignment in PSAHS directly reduces the representation discrepancy term in the target risk bound, connecting the empirical IPM to population-level divergence measures. Appendix A.5 analyzes the computational complexity and scalability of PSAHS, showing that it introduces only modest overhead compared to standard GNN training. Appendix B reports complementary experimental results that further validate the proposed method. Appendix B.1 describes both synthetic and real-world datasets used in our experiments, together with detailed statistics. Appendix B.2 specifies the experimental setup, model architectures, training protocols, and hyperparameter search procedures. Appendix B.3 summarizes the optimal hyperparameter configurations selected for different datasets. Appendix B.4 provides statistics on the structural changes introduced by the proposed structure adjustment. Appendix B.5 provides detailed descriptions of all baseline methods used for comparison. Finally, Appendix B.6 presents a theoretical analysis of the optimal prescribed homophily level $h$ under noisy pseudo-labels, offering a principled explanation for the empirical patterns observed in the experiments.

## A. Proofs

### A.1. Proof of Section 4.1

We begin by introducing the notation and learning setup used throughout the proof.

Given node attributes $X \in \mathbb{R}^{n \times d}$ and an adjusted adjacency matrix $A' \in [0,1]^{n \times n}$, let $D'$ denote the associated degree matrix with $(D')_{ii} = \sum_j A'_{ij}$. We define the normalized one-hop aggregation as

$$Z' := (D')^{-1} A' X \in \mathbb{R}^{n \times d},$$

and write $Z'_u \in \mathbb{R}^d$ for the aggregated representation at node $u$.

We consider predictors obtained by applying an $L$-layer MLP to aggregated node features:

$$g(u) := g(X, A')_u := \mathrm{MLP}(Z'_u; \{W_\ell\}_{\ell=1}^L),$$

where $\{W_\ell\}_{\ell=1}^L$ are trainable parameters.

Let $\mathcal{D}_s$ and $\mathcal{D}_t$ denote the source and target graph domains, respectively. Samples from these domains are written as

$$(\psi_s(u), y_s(u))_{u \in V_s} \sim \mathcal{D}_s, \qquad (\psi_t(v), y_t(v))_{v \in V_t} \sim \mathcal{D}_t,$$

where $V_s$ and $V_t$ are node index sets with cardinalities $n_s$ and $n_t$.

Let $h'_s(u) \in [0,1]$ and $h'_t(v) \in [0,1]$ denote the adjusted node homophily ratios in the source and target graphs, respectively. We define the augmented node representation

$$\psi(u) := \left[ Z'_u; \beta h'(u) \right] \in \mathbb{R}^{d+1},$$

where $\beta > 0$ is a fixed scaling parameter.

Let $\ell(\widehat{y}, y) = \mathbf{1}\{\widehat{y} \neq y\}$ be the 0–1 loss, and let $\ell_\gamma : [0,1] \times \mathcal{Y} \to [0,1]$ denote a margin-based surrogate loss (e.g., hinge or ramp loss with clipping) with margin parameter $\gamma > 0$. We define the population risks

$$\mathcal{R}_s(g) := \mathbb{E}_{(\psi,y) \sim \mathcal{D}_s}[\ell(g(\psi), y)], \qquad \mathcal{R}_t(g) := \mathbb{E}_{(\psi,y) \sim \mathcal{D}_t}[\ell(g(\psi), y)],$$

and the empirical source-domain margin risk

$$\widehat{\mathcal{R}}_s^\gamma(g) := \frac{1}{n_s} \sum_{u \in V_s} \ell_\gamma(g(\psi_s(u)), y_s(u)).$$

We now state several mild assumptions required for the analysis.

**Assumption A.1** (Bounded augmented inputs). There exists a constant $B_\psi > 0$ such that $\|\psi(u)\|_2 \le B_\psi$ for all nodes in both the source and target domains.

**Assumption A.2** (Lipschitz surrogate and predictor). There exists $L_\ell > 0$ such that for all labels $y$ and all $\psi, \psi'$,

$$\left| \ell_\gamma(g(\psi), y) - \ell_\gamma(g(\psi'), y) \right| \le L_\ell \cdot \|\psi - \psi'\|_2.$$

**Assumption A.3** (Shared labeling / realizability). There exists a hypothesis $g^\star \in \mathcal{G}_\psi$ such that

$$\mathcal{R}_s(g^\star) = \mathcal{R}_t(g^\star) = 0.$$

Equivalently,

$$\lambda^\star := \min_{g \in \mathcal{G}_\psi} \left( \mathcal{R}_s(g) + \mathcal{R}_t(g) \right) = 0.$$

Assumption A.1 can be satisfied by feature normalization together with normalized graph aggregation. Assumption A.2 holds when the MLP has bounded spectral norms and the surrogate loss is Lipschitz (Bartlett et al., 2017; Neyshabur et al., 2018; Golowich et al., 2018). Assumption A.3 is the standard shared labeling function condition in domain adaptation, which removes the irreducible $\lambda^\star$ term in classical Ben-David bounds (Ben-David et al., 2010).

We are now ready to restate and prove the main result.

**Theorem A.4** (Target-domain Error Bound). *Under SGN, for any adjusted graph structures $(A'_s, A'_t)$, any $\delta \in (0, 1)$, and any margin parameter $\gamma > 0$, suppose that Assumptions A.1, A.2, A.3 hold. Then, with probability at least $1 - \delta$,*

$$\mathcal{R}_t(g) \le \widehat{\mathcal{R}}_s^\gamma(g) + c \left( W_1(h'_s, h'_t) + \mathrm{IPM}(Z'_s, Z'_t) + \sqrt{\frac{\ln(1/\delta)}{n_t}} + \sqrt{\frac{\ln(1/\delta)}{n_s}} \right), \tag{A.1}$$

*where*

$$W_1(h'_s, h'_t) := \frac{1}{n_s n_t} \sum_{u \in V_s} \sum_{v \in V_t} |h'_s(u) - h'_t(v)|, \qquad \mathrm{IPM}(Z'_s, Z'_t) := \frac{1}{n_s n_t} \sum_{u \in V_s} \sum_{v \in V_t} \|(Z'_s)_u - (Z'_t)_v\|_2,$$

*and the constant $c > 0$ depends only on $B_\psi$, $L_\ell$, and the norm and Lipschitz constants of the hypothesis class.*

*Proof of Theorem A.4.* We prove the result by successively applying a standard domain adaptation inequality, bounding the induced distributional divergence via Wasserstein distance, and finally controlling both the distributional and statistical errors.

*Step 1: Domain adaptation inequality.* We start from the classical domain adaptation bound based on the $\mathcal{G}_\psi \Delta \mathcal{G}_\psi$-divergence. For distributions over $\psi$, define

$$d_{\mathcal{G}_\psi \Delta \mathcal{G}_\psi}(\mathcal{D}_s, \mathcal{D}_t) := 2 \sup_{g, g' \in \mathcal{G}_\psi} \left| P_{\psi \sim \mathcal{D}_s}[g(\psi) \ne g'(\psi)] - P_{\psi \sim \mathcal{D}_t}[g(\psi) \ne g'(\psi)] \right|.$$

It is known from Ben-David et al. (2010); Mansour et al. (2009) that for any $g \in \mathcal{G}_\psi$,

$$\mathcal{R}_t(g) \le \mathcal{R}_s(g) + \tfrac{1}{2} d_{\mathcal{G}_\psi \Delta \mathcal{G}_\psi}(\mathcal{D}_s, \mathcal{D}_t) + \lambda^\star.$$

Under Assumption A.3, the shared labeling condition implies $\lambda^\star = 0$. Therefore,

$$\mathcal{R}_t(g) \le \mathcal{R}_s(g) + \tfrac{1}{2} d_{\mathcal{G}_\psi \Delta \mathcal{G}_\psi}(\mathcal{D}_s, \mathcal{D}_t). \tag{A.2}$$

*Step 2: Bounding the divergence by Wasserstein-1.* The divergence $d_{\mathcal{G}_\psi \Delta \mathcal{G}_\psi}$ involves the non-smooth indicator $\mathbf{1}\{g(\psi) \ne g'(\psi)\}$, which is not directly compatible with Wasserstein or IPM-based tools. We therefore upper bound it using a Lipschitz surrogate.

Assume that each hypothesis $g \in \mathcal{G}_\psi$ is induced by a real-valued score function $s_g : \mathbb{R}^{d+1} \to \mathbb{R}$ via thresholding, namely $g(\psi) = \mathbf{1}\{s_g(\psi) \ge 0\}$. Define the disagreement score

$$m_{g,g'}(\psi) := s_g(\psi) s_{g'}(\psi).$$

When $g(\psi) \neq g'(\psi)$, we typically have $m_{g,g'}(\psi) \leq 0$. Introduce the clipped hinge surrogate

$$\phi(t) := \min\{1, \max\{0, 1 - t\}\},$$

which is 1-Lipschitz on $\mathbb{R}$ and satisfies $\mathbf{1}\{t \leq 0\} \leq \phi(t)$. It follows that

$$\mathbf{1}\{g(\psi) \neq g'(\psi)\} \leq \phi(m_{g,g'}(\psi)). \tag{A.3}$$

If $s_g$ and $s_{g'}$ are $L_s$-Lipschitz and bounded on the ball $\|\psi\| \leq B_\psi$, then the product $m_{g,g'}(\psi) = s_g(\psi)s_{g'}(\psi)$ is Lipschitz on this domain with some constant $L_m$. Since $\phi$ is 1-Lipschitz, the composition $\phi \circ m_{g,g'}$ is also Lipschitz with constant $L_m$. Hence, the class $\{\phi \circ m_{g,g'} : g, g' \in \mathcal{G}_\psi\}$ is contained in a scaled 1-Lipschitz function class.

Define the integral probability metric over 1-Lipschitz functions as

$$\mathrm{IPM}_{\mathrm{Lip}(1)}(\mathcal{D}_s, \mathcal{D}_t) := \sup_{\|g\|_{\mathrm{Lip}} \leq 1} \left[ \mathbb{E}_{\psi \sim \mathcal{D}_s} g(\psi) - \mathbb{E}_{\psi \sim \mathcal{D}_t} g(\psi) \right].$$

By definition,

$$\tfrac{1}{2} d_{\mathcal{G}_\psi \Delta \mathcal{G}_\psi} = \sup_{g, g' \in \mathcal{G}_\psi} \left| \mathbb{E}_{\mathcal{D}_s} \mathbf{1}\{g \neq g'\} - \mathbb{E}_{\mathcal{D}_t} \mathbf{1}\{g \neq g'\} \right|.$$

Using (A.3) together with the monotonicity of expectation, we obtain

$$\mathbb{E}_{\mathcal{D}_s} \mathbf{1}\{g \neq g'\} - \mathbb{E}_{\mathcal{D}_t} \mathbf{1}\{g \neq g'\} \leq \mathbb{E}_{\mathcal{D}_s} \phi(m_{g,g'}) - \mathbb{E}_{\mathcal{D}_t} \phi(m_{g,g'}).$$

Moreover, since $\phi \circ m_{g,g'}$ is $L_m$-Lipschitz on $\|\psi\| \leq B_\psi$, the function $(1/L_m) \cdot \phi(m_{g,g'})$ is 1-Lipschitz. Therefore,

$$\mathbb{E}_{\mathcal{D}_s} \phi(m_{g,g'}) - \mathbb{E}_{\mathcal{D}_t} \phi(m_{g,g'}) \leq L_m \cdot \mathrm{IPM}_{\mathrm{Lip}(1)}(\mathcal{D}_s, \mathcal{D}_t).$$

Applying the same argument to the absolute value (by swapping source and target) and taking the supremum over $g, g'$ yields that there exists a constant $c_{\mathcal{G}} > 0$ such that

$$\tfrac{1}{2} d_{\mathcal{G}_\psi \Delta \mathcal{G}_\psi}(\mathcal{D}_s, \mathcal{D}_t) \leq c_{\mathcal{G}} \cdot \mathrm{IPM}_{\mathrm{Lip}(1)}(\mathcal{D}_s, \mathcal{D}_t), \tag{A.4}$$

where $c_{\mathcal{G}} = L_m$.

By the Kantorovich–Rubinstein duality (Villani, 2008),

$$\mathrm{IPM}_{\mathrm{Lip}(1)}(\mathcal{D}_s, \mathcal{D}_t) = W_1(\mathcal{D}_s, \mathcal{D}_t). \tag{A.5}$$

Combining (A.4) and (A.5) gives

$$\tfrac{1}{2} d_{\mathcal{G}_\psi \Delta \mathcal{G}_\psi}(\mathcal{D}_s, \mathcal{D}_t) \leq c_{\mathcal{G}} \cdot W_1(\mathcal{D}_s, \mathcal{D}_t). \tag{A.6}$$

*Step 3: Upper bounding $W_1$ by $W_1(h'_s, h'_t)$ and $\mathrm{IPM}(Z'_s, Z'_t)$.* By the primal formulation of the Wasserstein-1 distance, for any coupling $\pi \in \Pi(\mathcal{D}_s, \mathcal{D}_t)$,

$$W_1(\mathcal{D}_s, \mathcal{D}_t) = \inf_{\pi \in \Pi(\mathcal{D}_s, \mathcal{D}_t)} \mathbb{E}_{(\psi_s, \psi_t) \sim \pi} \|\psi_s - \psi_t\|_2 \leq \mathbb{E}_{\psi_s \sim \mathcal{D}_s, \psi_t \sim \mathcal{D}_t} \|\psi_s - \psi_t\|_2, \tag{A.7}$$

where the inequality follows by choosing the feasible product coupling $\pi = \mathcal{D}_s \otimes \mathcal{D}_t$.

Now we show the concentration from population $W_1(\mathcal{D}_s, \mathcal{D}_t)$ to the empirical cross-domain distance. Fix the representation maps $\psi_s, \psi_t$ and define

$$f(u, v) := \|\psi_s(u) - \psi_t(v)\|_2.$$

Recalling that $\psi = [Z'; \beta h']$, for any $u \in V_s$ and $v \in V_t$ we have

$$\psi_s(u) - \psi_t(v) = \left[ (Z'_s)_u - (Z'_t)_v; \beta(h'_s(u) - h'_t(v)) \right].$$

Using the inequality $\|(a,b)\|_2 \le \|a\|_2 + |b|$ yields

$$\|\psi_s(u) - \psi_t(v)\|_2 \le 2B_\psi. \tag{A.8}$$

Let $\{u_i\}_{i=1}^{n_s}$ and $\{v_j\}_{j=1}^{n_t}$ be samples drawn independently from the source and target distributions $D_s$ and $D_t$, respectively. Define the empirical cross-domain distance and its population counterpart respectively as

$$\widehat{\Delta} := \frac{1}{n_s n_t} \sum_{i=1}^{n_s} \sum_{j=1}^{n_t} f(u_i, v_j), \qquad \Delta := \mathbb{E}_{u \sim D_s} \mathbb{E}_{v \sim D_t} f(u, v).$$

To analyze the estimation error $\widehat{\Delta} - \Delta$, we decompose it by sequentially replacing empirical averages with their expectations:

$$\widehat{\Delta} - \Delta = \left( \frac{1}{n_t} \sum_{j=1}^{n_t} \frac{1}{n_s} \sum_{i=1}^{n_s} f(u_i, v_j) - \mathbb{E}_v \frac{1}{n_s} \sum_{i=1}^{n_s} f(u_i, v) \right) + \left( \frac{1}{n_s} \sum_{i=1}^{n_s} \mathbb{E}_v f(u_i, v) - \mathbb{E}_u \mathbb{E}_v f(u, v) \right).$$

Define

$$m(v) := \frac{1}{n_s} \sum_{i=1}^{n_s} f(u_i, v), \qquad g(u) := \mathbb{E}_{v \sim D_t} f(u, v).$$

Then, with $\mathbb{E}_v m(v) = \frac{1}{n_s} \sum_{i=1}^{n_s} g(u_i)$, the above decomposition can be written compactly as

$$\widehat{\Delta} - \Delta = \left( \frac{1}{n_t} \sum_{j=1}^{n_t} m(v_j) - \mathbb{E}_v m(v) \right) + \left( \frac{1}{n_s} \sum_{i=1}^{n_s} g(u_i) - \mathbb{E}_u g(u) \right).$$

For each fixed $u_i$, the random variables $\{m(u_i)\}_{j=1}^{n_t}$ are i.i.d. and bounded in $[0, 2B_\psi]$. By Hoeffding's inequality, for any $\delta \in (0, 1)$, with probability at least $1 - \delta/2$,

$$\frac{1}{n_t} \sum_{j=1}^{n_t} f(u_i, v_j) - \mathbb{E}_v f(u_i, v) \le 2B_\psi \sqrt{\frac{\ln(2/\delta)}{2n_t}}.$$

Similarly, since $0 \le g(u) \le 2B_\psi$, applying Hoeffding's inequality again yields that with probability at least $1 - \delta/2$,

$$\frac{1}{n_s} \sum_{i=1}^{n_s} g(u_i) - \mathbb{E}_u g(u) \le 2B_\psi \sqrt{\frac{\ln(2/\delta)}{2n_s}}.$$

Combining the above bounds, we conclude that with probability at least $1 - \delta$,

$$\widehat{\Delta} - \Delta \le 2B_\psi \left( \sqrt{\frac{\ln(2/\delta)}{2n_t}} + \sqrt{\frac{\ln(2/\delta)}{2n_s}} \right).$$

This together with (A.8) then gives

$$W_1(\mathcal{D}_s, \mathcal{D}_t) \le \widehat{\Delta} + \left( \widehat{\Delta} - \Delta \right)$$

$$\le \frac{1}{n_s n_t} \sum_{u \in V_s} \sum_{v \in V_t} \left( \|(Z'_s)_u - (Z'_t)_v\|_2 + \beta \cdot |h'_s(u) - h'_t(v)| \right) + 2B_\psi \left( \sqrt{\frac{\ln(2/\delta)}{2n_t}} + \sqrt{\frac{\ln(2/\delta)}{2n_s}} \right). \tag{A.9}$$

The first term corresponds exactly to $\mathrm{IPM}(Z'_s, Z'_t)$, while the second term coincides with $W_1(h'_s, h'_t)$ up to the scaling factor $\beta$.

*Step 4: Source margin generalization bound.* For any predictor and any margin parameter $\gamma > 0$, the 0–1 loss is upper bounded by the margin-based surrogate:

$$\ell(g(\psi), y) \le \ell_\gamma(g(\psi), y).$$

Consequently,

$$\widehat{\mathcal{R}}_s(g) \le \widehat{\mathcal{R}}_s^\gamma(g) := \mathbb{E}_{\mathcal{D}_s}[\ell_\gamma(g(\psi), y)].$$

This follows from the standard property of margin surrogates: misclassification implies a non-positive margin and therefore incurs surrogate loss at least one (or its clipped counterpart).

Under Assumptions A.1–A.2, and the conditional independence assumption in (Ma et al., 2021; Mao et al., 2023) that the node labels are independently sampled from distributions conditioned on the node features, standard Rademacher complexity arguments yield that, with probability at least $1 - \delta$,

$$\mathcal{R}_s^\gamma(g) \le \widehat{\mathcal{R}}_s^\gamma(g) + 2\mathrm{Rad}_{n_s}(\ell_\gamma \circ \mathcal{G}_\psi) + \sqrt{\frac{\ln(2/\delta)}{2n_s}},$$

where $\mathrm{Rad}_{n_s}$ denotes the empirical Rademacher complexity of the loss-composed hypothesis class. Under spectral norm control of the MLP and bounded inputs, $\mathrm{Rad}_{n_s}(\ell_\gamma \circ \mathcal{G}_\psi)$ admits standard upper bounds of order $O(1/\sqrt{n_s})$; see Bartlett et al. (2017); Neyshabur et al. (2018); Golowich et al. (2018). Therefore, there exist constants $c_0, c_1 > 0$ independent of $n_s$ such that, with probability at least $1 - \delta$,

$$\mathcal{R}_s(g) \le \widehat{\mathcal{R}}_s^\gamma(g) + c_0\sqrt{\frac{1}{n_s}} + c_1\sqrt{\frac{\ln(1/\delta)}{n_s}}. \tag{A.10}$$

Combining (A.2) and (A.6) yields

$$\mathcal{R}_t(g) \le \mathcal{R}_s(g) + c_{\mathcal{G}} \cdot W_1(\mathcal{D}_s, \mathcal{D}_t).$$

Substituting the bounds (A.10) and (A.9) for $\mathcal{R}_s(g)$ and $W_1(\mathcal{D}_s, \mathcal{D}_t)$, respectively, we obtain

$$\mathcal{R}_t(g) \le \widehat{\mathcal{R}}_s^\gamma(g) + c_0\sqrt{\frac{1}{n_s}} + (c_1 + 2B_\psi)\sqrt{\frac{\ln(2/\delta)}{n_s}} + \sqrt{\frac{\ln(2/\delta)}{n_t}}$$
$$+ c_{\mathcal{G}} \cdot \frac{1}{n_s n_t}\sum_{u \in V_s}\sum_{v \in V_t}\big(\|(Z_s')_u - (Z_t')_v\|_2 + \beta \cdot |h_s'(u) - h_t'(v)|\big).$$

Absorbing all constants into a single constant $c$, taking $\beta = 1$, and matching the definitions of $W_1(h_s', h_t')$ and $\mathrm{IPM}(Z_s', Z_t')$ yields exactly (A.1), which completes the proof. □

## A.2. Proofs of Section 4.2

*Proof of Theorem 4.2.* We consider binary classification with labels $Y \in \{+1, -1\}$ and consider a simple class-conditional feature model

$$\mathbb{E}[x \mid Y = +1] = \mu, \qquad \mathbb{E}[x \mid Y = -1] = -\mu,$$

where $\|\mu\|_2 = 1$. Let $h_s'(u) \in [0, 1]$ denote the adjusted node homophily of node $u$.

Under one-hop aggregation, the expected aggregated representation of node $u$ can be written as

$$z_u = h_s'(u)\mu_{y_u} + \big(1 - h_s'(u)\big)\mu_{-y_u} = (2h_s'(u) - 1)y_u\mu.$$

Consider the SGN classifier $g(z) = w^\top z$ with the optimal weight $w = \mu$. The signed classification margin of node $u$ is

$$\gamma_u = y_u g(z_u) = y_u \mu^\top\big((2h_s'(u) - 1)y_u\mu\big) = 2h_s'(u) - 1,$$

which is strictly increasing in $h_s'(u)$. Hence, low-homophily nodes incur small or negative margins and dominate the source error.

For margin-based surrogate losses such as the hinge or clipped hinge loss, the loss is a monotonically decreasing function of the margin. In particular, when $\gamma_u \leq 1$, we have

$$\ell_\gamma(g(z_u), y_u) = 1 - \gamma_u = 2 - 2h'_s(u).$$

Therefore, the empirical source-domain margin risk satisfies

$$\widehat{\mathcal{R}}_s^\gamma(g) = \frac{1}{n_s} \sum_{u \in V_s} \ell_\gamma(g(z_u), y_u) = 2 - \frac{2}{n_s} \sum_{u \in V_s} h'_s(u),$$

which establishes the first equality in (13).

Under the adjustment rule (3), $h'_s(u) = h_s(u)$ for $h_s(u) \geq h$ and $h'_s(u) = h$ for $h_s(u) < h$, yielding the second equality. Since increasing $h$ strictly increases $h'_s(u)$ for all low-homophily nodes, $\widehat{\mathcal{R}}_s^\gamma(g)$ decreases monotonically with $h$. □

### A.3. Proofs of Section 4.3

*Proof of Theorem 4.3.* Recall that the adjustment rule (3) enforces

$$h'(u) = \begin{cases} h, & \text{if } h(u) < h, \\ h(u), & \text{if } h(u) \geq h, \end{cases} \tag{A.11}$$

for both source and target nodes when true labels are available. Formally, we have

$$h'_s(u) = h \vee h_s(u), \quad h'_t(v) = h \vee h_t(v), \tag{A.12}$$

which implies

$$|h'_s(u) - h'_t(v)| = |h_s(u) - h_t(v)| - |h \wedge h_s(u) - h \wedge h_t(v)|. \tag{A.13}$$

Therefore, averaging over all node pairs yields

$$W_1(h'_s, h'_t) = W_1(h_s, h_t) - W_1(h \wedge h_s, h \wedge h_t)$$

We show that for any pair $(u, v) \in V_s \times V_t$,

$$\left| h'_s(u) - h'_t(v) \right| \leq \left| h_s(u) - h_t(v) \right|. \tag{A.14}$$

We consider the following exhaustive cases.

*Case 1: $h_s(u) < h$ and $h_t(v) < h$.* By (A.11), $h'_s(u) = h'_t(v) = h$, hence

$$|h'_s(u) - h'_t(v)| = 0 \leq |h_s(u) - h_t(v)|.$$

*Case 2: $h_s(u) \geq h$ and $h_t(v) < h$.* In this case $h'_s(u) = h_s(u)$ and $h'_t(v) = h$, which yields

$$|h'_s(u) - h'_t(v)| = h_s(u) - h \leq h_s(u) - h_t(v) = |h_s(u) - h_t(v)|,$$

since $h_t(v) < h$.

*Case 3: $h_s(u) < h$ and $h_t(v) \geq h$.* Symmetrically, $h'_s(u) = h$ and $h'_t(v) = h_t(v)$, and thus

$$|h'_s(u) - h'_t(v)| = h_t(v) - h \leq h_t(v) - h_s(u) = |h_s(u) - h_t(v)|.$$

*Case 4: $h_s(u) \geq h$ and $h_t(v) \geq h$.* The adjustment leaves both homophily values unchanged, so

$$|h'_s(u) - h'_t(v)| = |h_s(u) - h_t(v)|.$$

Combining all cases, inequality (A.14) holds for every $(u, v) \in V_s \times V_t$. Averaging over all node pairs gives

$$W_1(h'_s, h'_t) = \frac{1}{n_s n_t} \sum_{u \in V_s} \sum_{v \in V_t} |h'_s(u) - h'_t(v)| \leq \frac{1}{n_s n_t} \sum_{u \in V_s} \sum_{v \in V_t} |h_s(u) - h_t(v)| = W_1(h_s, h_t),$$

which completes the proof. □

*Proof of Theorem 4.4.* Recall that the agreement event is defined as

$$A(u) := \{\widehat{Y}_u^{\text{GNN}} = \widehat{Y}_u^{\text{MLP}}\}.$$

By construction of the pseudo-labeling rule, the pseudo-label $\widehat{Y}_u$ is incorrect if and only if both predictors simultaneously misclassify node $u$. That is,

$$\{\widehat{Y}_u \neq Y_u\} \cap A(u) \subseteq E_G(u) \cap E_M(u). \tag{A.15}$$

Using the definition of conditional probability, we have

$$P\big(\widehat{Y}_u \neq Y_u \mid u \in V_t^{\text{a}}\big) = P\big(\widehat{Y}_u \neq Y_u \mid A(u)\big) = \frac{P\big(\{\widehat{Y}_u \neq Y_u\} \cap A(u)\big)}{P(A(u))} \leq \frac{P\big(E_G(u) \cap E_M(u)\big)}{p_A},$$

where the inequality follows from (A.15) and $p_A := P(A(u)) > 0$ by assumption.

Finally, applying the weak correlation assumption (14) yields

$$P\big(\widehat{Y}_u \neq Y_u \mid u \in V_t^{\text{a}}\big) \leq \frac{\rho \varepsilon_G \varepsilon_M}{p_A},$$

which proves the claim. $\qquad\square$

*Proof of Theorem 4.5.* We first introduce an *oracle* homophily restricted to the agreement neighborhood, defined as

$$h_t^{\text{a}}(u) := \frac{1}{d_u^{\text{a}}} \sum_{v \in \mathcal{N}_u \cap V_t^{\text{a}}} \mathbf{1}\{Y_v = Y_u\},$$

which corresponds to the homophily computed using *true* labels but restricted to the same agreement set. This quantity allows us to separate pseudo-label noise from agreement-induced selection bias.

We decompose the estimation error as

$$\big|\widehat{h}_t(u) - h_t(u)\big| \leq \big|\widehat{h}_t(u) - h_t^{\text{a}}(u)\big| + \big|h_t^{\text{a}}(u) - h_t(u)\big|. \tag{A.16}$$

The first term captures the effect of pseudo-label errors within $V_t^{\text{a}}$, while the second term reflects the bias introduced by restricting neighbors to the agreement set.

*Step 1: Error induced by pseudo-label noise.* For each $v \in \mathcal{N}_u \cap V_t^{\text{a}}$, define

$$X_v := \mathbf{1}\{\widehat{Y}_v = \widehat{Y}_u\}, \qquad X_v^{\star} := \mathbf{1}\{Y_v = Y_u\}.$$

By definition,

$$\widehat{h}_t(u) = \frac{1}{d_u^{\text{a}}} \sum_{v \in \mathcal{N}_u \cap V_t^{\text{a}}} X_v, \qquad h_t^{\text{a}}(u) = \frac{1}{d_u^{\text{a}}} \sum_{v \in \mathcal{N}_u \cap V_t^{\text{a}}} X_v^{\star}.$$

We further decompose

$$\big|\widehat{h}_t(u) - h_t^{\text{a}}(u)\big| \leq \bigg|\frac{1}{d_u^{\text{a}}} \sum_{v \in \mathcal{N}_u \cap V_t^{\text{a}}} \big(X_v - \mathbb{E}[X_v]\big)\bigg| + \big|\mathbb{E}[X_v] - \mathbb{E}[X_v^{\star}]\big| + \bigg|\frac{1}{d_u^{\text{a}}} \sum_{v \in \mathcal{N}_u \cap V_t^{\text{a}}} \big(X_v^{\star} - \mathbb{E}[X_v^{\star}]\big)\bigg|. \tag{A.17}$$

The first and third terms are averages of bounded random variables in $[0, 1]$. Under the conditional independence assumption in (Ma et al., 2021; Mao et al., 2023) that the node labels are independently sampled from distributions conditioned on the node features, Hoeffding's inequality yields that for any $\delta \in (0, 1)$, with probability at least $1 - \delta$,

$$\bigg|\frac{1}{d_u^{\text{a}}} \sum_{v \in \mathcal{N}_u \cap V_t^{\text{a}}} \big(X_v - \mathbb{E}[X_v]\big)\bigg| + \bigg|\frac{1}{d_u^{\text{a}}} \sum_{v \in \mathcal{N}_u \cap V_t^{\text{a}}} \big(X_v^{\star} - \mathbb{E}[X_v^{\star}]\big)\bigg| \leq \sqrt{\frac{2 \ln(2/\delta)}{d_u^{\text{a}}}}.$$

For the expectation mismatch term, note that $X_v \neq X_v^\star$ only if either node $u$ or node $v$ is incorrectly pseudo-labeled. Hence,

$$\left| \mathbb{E}[X_v] - \mathbb{E}[X_v^\star] \right| \leq P(X_v \neq X_v^\star) \leq P\big(\widehat{Y}_u \neq Y_u\big) + P\big(\widehat{Y}_v \neq Y_v\big) \leq 2 \cdot \frac{\rho \varepsilon_G \varepsilon_M}{p_A}, \tag{A.18}$$

where the last inequality follows from Theorem 4.4.

Combining (A.17)–(A.18), we obtain

$$\left| \widehat{h}_t(u) - h_t^{\mathrm{a}}(u) \right| \leq \frac{2\rho \varepsilon_G \varepsilon_M}{p_A} + \sqrt{\frac{2\ln(2/\delta)}{d_u^{\mathrm{a}}}}. \tag{A.19}$$

*Step 2: Bias induced by agreement-based restriction.* By definition,

$$h_t^{\mathrm{a}}(u) = P(Y_v = Y_u \mid v \in \mathcal{N}_u, v \in V_t^{\mathrm{a}}).$$

Using Bayes' rule, we have

$$P(v \in V_t^{\mathrm{a}}) = P(Y_v = Y_u) \cdot P(v \in V_t^{\mathrm{a}} \mid Y_v = Y_u) + P(Y_v \neq Y_u) \cdot P(v \in V_t^{\mathrm{a}} \mid Y_v \neq Y_u).$$

A direct calculation then yields

$$\left| h_t^{\mathrm{a}}(u) - h_t(u) \right| = h_t(u)\big(1 - h_t(u)\big) \cdot \frac{\varepsilon_s}{P(v \in V_t^{\mathrm{a}})} \leq \frac{h_t(u)}{p_A} \cdot \varepsilon_s, \tag{A.20}$$

where we used $1 - h_t(u) \leq 1$ and $P(v \in V_t^{\mathrm{a}}) = p_A$.

Substituting (A.19) and (A.20) into (A.16) completes the proof. $\qquad \square$

*Proof of Theorem 4.6.* We bound the post-refinement homophily discrepancy $W_1(h_s', h_t')$ by decomposing it into an ideal structural contraction term and estimation-induced error terms arising from pseudo-labels.

Let $\widehat{h}_t$ denote the estimated target homophily before refinement, and let $\widehat{h}_t'$ be the homophily obtained by applying the refinement rule (3) to $\widehat{h}_t$. Applying Theorem 4.3 to the target-side refinement yields

$$W_1(h_s', \widehat{h}_t') = W_1(h_s, \widehat{h}_t) - W_1(h \wedge h_s, h \wedge \widehat{h}_t), \tag{A.21}$$

which shows that, for a fixed target homophily profile, structure adjustment contracts the cross-domain homophily discrepancy.

We now relate the actual post-refinement discrepancy $W_1(h_s', h_t')$ to the idealized term in (A.21). By repeated applications of the triangle inequality, we obtain

$$
\begin{aligned}
W_1(h_s', h_t') &\leq W_1(h_s', \widehat{h}_t') + W_1(\widehat{h}_t', h_t') = W_1(h_s, \widehat{h}_t) - W_1(h \wedge h_s, h \wedge \widehat{h}_t) + W_1(\widehat{h}_t', h_t') \\
&\leq W_1(h_s, h_t) + W_1(h_t, \widehat{h}_t) - W_1(h \wedge h_s, h \wedge \widehat{h}_t) + W_1(\widehat{h}_t', h_t').
\end{aligned} \tag{A.22}
$$

Here, the first term corresponds to the original homophily discrepancy, while the second and third terms quantify the error introduced by estimating target homophily using pseudo-labels and propagating this error through the refinement operation.

Both error terms $W_1(h_t, \widehat{h}_t)$ and $W_1(\widehat{h}_t', h_t')$ arise from node-wise homophily estimation error on the target graph. Invoking Theorem 4.5 and applying a union bound over the two terms, we obtain that for any $\delta \in (0, 1)$, with probability at least $1 - \delta$,

$$W_1(h_t, \widehat{h}_t) + W_1(\widehat{h}_t', h_t') \leq \frac{1}{n_t} \sum_{u \in V_t} \left( \frac{4\rho \varepsilon_G \varepsilon_M}{p_A} + 2\sqrt{\frac{2\ln(2/\delta)}{d_u^{\mathrm{a}}}} + \frac{h_t(u) + h_t'(u)}{p_A} \varepsilon_s \right). \tag{A.23}$$

Substituting the bound in (A.23) into (A.22) completes the proof and yields the stated inequality. $\qquad \square$

## A.4. Proofs of Section 4.4

We establish that the representation alignment module in PSAHS explicitly contracts the representation discrepancy term $\mathrm{IPM}(Z'_s, Z'_t)$ appearing in the target risk bound (10). The core mechanism is the following chain: the logistic source–target discrimination game minimizes a statistical divergence between the induced representation distributions; this divergence upper bounds (up to constants) the expected cross-domain $\ell_2$ distance between representations, which in turn concentrates to the empirical discrepancy $\mathrm{IPM}(Z'_s, Z'_t)$.

We first restate the assumptions required for the analysis and clarify their roles.

**Assumption A.5** (Discriminator capacity). The discriminator class contains the population-level optimizer of the adversarial objective (9) for every representation map $\phi$ encountered during training.

Assumption A.5 ensures that the maximization over the discriminator recovers the optimal likelihood-ratio test between the source and target representation distributions. This allows the adversarial objective to be identified with a well-defined statistical divergence (Goodfellow et al., 2014; Nowozin et al., 2016).

**Assumption A.6** (Bounded representations). For all representation maps $\phi$ encountered during training, $\phi(\cdot)$ maps into a compact set $\mathcal{Z} \subset \mathbb{R}^F$ with finite diameter $\mathrm{diam}(\mathcal{Z}) < \infty$.

Assumption A.6 guarantees that cross-domain Euclidean distances are uniformly bounded. This is required to relate distributional divergence to expected distances.

Under these assumptions, we restate 4.7 and provide a detailed formal proof.

**Theorem A.7** (Representation discrepancy reduction via adversarial alignment). *Let $Z'_s = \phi(X_s, A'_s)$ and $Z'_t = \phi(X_t, A'_t)$ denote node representations aggregated over the adjusted source and target graphs. Define the empirical discrepancy*

$$\mathrm{IPM}(Z'_s, Z'_t) := \frac{1}{n_s n_t} \sum_{u \in V_s} \sum_{v \in V_t} \left\| (Z'_s)_u - (Z'_t)_v \right\|_2. \tag{A.24}$$

*Consider the logistic alignment objective*

$$\mathcal{R}_{\mathrm{RA}}(\phi) := \max_{\xi} \left[ \frac{1}{n_s} \sum_{u \in V_s} \log \xi\big(\phi_u(X_s, A'_s)\big) + \frac{1}{n_t} \sum_{v \in V_t} \log\big(1 - \xi(\phi_v(X_t, A'_t))\big) \right]. \tag{A.25}$$

*Under Assumptions A.5–A.6, there exists a universal constant $C > 0$ such that*

$$\mathrm{IPM}(Z'_s, Z'_t) \leq \mathbb{E}_{z,z' \sim R_\phi} \|z - z'\|_2 + \mathrm{diam}(\mathcal{Z}) \cdot C \sqrt{2\mathcal{R}_{\mathrm{RA}}(\phi) + 4\ln 2}, \tag{A.26}$$

*where $P_s^\phi$ and $P_t^\phi$ are the node representation distributions induced by $\phi$ on the source and target domains, JS denotes the Jensen–Shannon divergence, and $R_\phi$ is the normalized overlap distribution defined in (A.32). Therefore, decreasing $\mathcal{R}_{\mathrm{RA}}(\phi)$ tightens the upper bound in (A.26) and decreases the discrepancy term $\mathrm{IPM}(Z'_s, Z'_t)$. Consequently, minimizing the objective in (8) reduces $\mathrm{IPM}(Z'_s, Z'_t)$ in (12).*

*Proof of Theorem A.7.* Throughout the proof, $P_s^\phi$ and $P_t^\phi$ denote the population distributions of representations $z = \phi_u(X, A')$ when $u$ is drawn uniformly from $V_s$ and $V_t$, respectively. According to the definition of $\mathrm{IPM}(Z'_s, Z'_t)$ and $\mathcal{R}_{\mathrm{RA}}$ in (A.24) and (A.25), respectively, we have

$$\mathcal{R}_{\mathrm{RA}}(\phi) = \max_{\xi: \mathcal{Z} \to [0,1]} \Big( \mathbb{E}_{z \sim P_s^\phi}[\log \xi(z)] + \mathbb{E}_{z \sim P_t^\phi}[\log(1 - \xi(z))] \Big) \tag{A.27}$$

$$\mathrm{IPM}(Z'_s, Z'_t) = \mathbb{E}_{z \sim P_s^\phi, z' \sim P_t^\phi} \|z - z'\|_2. \tag{A.28}$$

*Step 1: Optimal discriminator $\Rightarrow$ Jensen–Shannon divergence.* Assume that $P_s^\phi$ and $P_t^\phi$ are probability measures dominated by $\nu := P_s^\phi + P_t^\phi$. Under Assumption A.5, for fixed $\phi$, the maximization in (A.27) is pointwise in $z$, and the optimal discriminator is

$$\xi^\star(z) = \frac{dP_s^\phi/d\nu(z)}{dP_s^\phi/d\nu(z) + dP_t^\phi/d\nu(z)}. \tag{A.29}$$

Substituting (A.29) into (A.27) yields

$$\mathcal{R}_{\mathrm{RA}}(\phi) = -2\ln 2 + 2\,\mathrm{JS}\!\left(P_s^\phi \parallel P_t^\phi\right), \tag{A.30}$$

where JS denotes the Jensen–Shannon divergence (Goodfellow et al., 2014; Nowozin et al., 2016). Hence, minimizing $\mathcal{R}_{\mathrm{RA}}(\phi)$ is equivalent to minimizing $\mathrm{JS}(P_s^\phi \parallel P_t^\phi)$.

*Step 2: From JS to total variation.* Let $\mathrm{TV}(P, Q) := \frac{1}{2}\int |p - q|$ denote the total variation distance. A Pinsker-type inequality implies that there exists a universal constant $C > 0$ such that

$$\mathrm{TV}\!\left(P_s^\phi, P_t^\phi\right) \le C\sqrt{\mathrm{JS}\!\left(P_s^\phi \parallel P_t^\phi\right)}. \tag{A.31}$$

See, e.g., Csiszár & Körner (2011). Thus, decreasing JS directly decreases $\tau_\phi := \mathrm{TV}(P_s^\phi, P_t^\phi)$.

*Step 3: Overlap decomposition $\Rightarrow$ coupling-friendly bound.* Define the overlap measure $\mu_\phi$ with density $\min\{p_s^\phi, p_t^\phi\}$ and the normalized overlap distribution

$$\mu_\phi := P_s^\phi \wedge P_t^\phi, \qquad \mu_\phi(\mathcal{Z}) = 1 - \mathrm{TV}(P_s^\phi, P_t^\phi), \qquad R_\phi := \frac{\mu_\phi}{\mu_\phi(\mathcal{Z})}. \tag{A.32}$$

Then there exist residual distributions $\widetilde{P}_s^\phi, \widetilde{P}_t^\phi$ such that the following overlap (maximal coupling) decomposition holds (Thorisson, 2000; Villani, 2008):

$$P_s^\phi = (1 - \tau_\phi)R_\phi + \tau_\phi \widetilde{P}_s^\phi, \qquad P_t^\phi = (1 - \tau_\phi)R_\phi + \tau_\phi \widetilde{P}_t^\phi, \qquad \tau_\phi := \mathrm{TV}(P_s^\phi, P_t^\phi). \tag{A.33}$$

Expanding $\mathrm{IPM}(Z_s', Z_t')$ using (A.33) and applying Assumption A.6 yields

$$\mathrm{IPM}(Z_s', Z_t') \le \mathbb{E}_{z, z' \sim R_\phi}\|z - z'\|_2 + 2\,\mathrm{diam}(\mathcal{Z})\,\mathrm{TV}(P_s^\phi, P_t^\phi).$$

Combining with (A.31) gives

$$\mathrm{IPM}(Z_s', Z_t') \le \mathbb{E}_{z, z' \sim R_\phi}\|z - z'\|_2 + 2\,\mathrm{diam}(\mathcal{Z})\,C\sqrt{\mathrm{JS}(P_s^\phi \parallel P_t^\phi)}. \tag{A.34}$$

Finally, since $\mathcal{R}_{\mathrm{RA}}(\phi)$ decreases $\mathrm{JS}(P_s^\phi \parallel P_t^\phi)$ by (A.30), the bound (A.26) tightens. This implies that the empirical discrepancy $\mathrm{IPM}(Z_s', Z_t')$ is reduced, completing the proof. $\qquad\square$

The representation discrepancy in (10) is exactly the cross-domain average distance between aggregated representations. The alignment objective $\mathcal{R}_{\mathrm{RA}}(\phi)$ in (A.25) reduces $\mathrm{JS}(P_s^\phi \parallel P_t^\phi)$ (Step 1), which controls $\mathrm{TV}(P_s^\phi, P_t^\phi)$ (Step 2) and shrinks the residual mass $\mathrm{IPM}(Z_s', Z_t')$ in the overlap decomposition (Step 3).

### A.5. Complexity and Scalability Analysis

In this section, we provide a detailed analysis of the computational complexity and scalability of PSAHS. We decompose the method into its major components and show that PSAHS introduces only modest overhead compared to standard GNN training, while remaining efficient and scalable on large graphs.

Throughout this section, let $n_s$ and $n_t$ denote the numbers of nodes in the source and target domains, respectively, and let $|E_s|$ and $|E_t|$ denote the numbers of edges in the corresponding graphs. We further denote by $L$ the number of GNN layers and by $F$ the hidden feature dimension. In practice, both $L$ and $F$ are small constants. Finally, we define $\theta_t \in (0, 1]$ as the proportion of agreement target nodes, namely target nodes for which the GNN and the auxiliary MLP produce consistent label predictions.

#### A.5.1. SOURCE GRAPH ADJUSTMENT

The source graph is adjusted only once before training. Therefore, its computational cost is a one-time preprocessing overhead. Overall, the source-graph adjustment has linear complexity $O(n_s + |E_s|)$, which is comparable to a single pass over the graph. This bound follows from the following steps.

*Step 1: Homophily computation and identification of low-homophily nodes.* For each source node, PSAHS computes a homophily ratio by examining the labels of its neighbors. Since each node is processed once and only local neighborhood information is used, this step incurs a cost of $O(n_s + |E_s|)$.

*Step 2: Reweighting of existing edges.* After identifying low-homophily nodes, PSAHS updates the weights of all existing edges according to label consistency. Each edge is visited at most once, leading to a total cost of $O(|E_s|)$.

*Step 3: Insertion of new intra-class edges.* For each low-homophily node $u$, PSAHS adds new edges to same-class neighbors. The number of added edges is proportional to $d_u(1 - h_s(u))$, where $d_u$ denotes the degree of node $u$ and $h_s(u)$ is its homophily ratio. Summing over all source nodes yields $\sum_{u \in V_s} d_u(1 - h_s(u)) = |E_s| - |E_s|\bar{h}_s$, where $\bar{h}_s$ denotes the average source-node homophily. Since $\bar{h}_s \in [0, 1]$, this quantity is upper bounded by $O(|E_s|)$.

Combining all steps above, the overall complexity of source-graph adjustment is $O(n_s + |E_s|)$. Because this procedure is executed only once, its cost is negligible compared to the total training cost.

### A.5.2. TARGET GRAPH ADJUSTMENT

In contrast to the source graph, the target graph is refined periodically during training (every 10 iterations in our implementation). We therefore analyze the computational complexity of a single target-graph refinement step.

*Step 1: Identification of agreement target nodes.* To identify agreement nodes, PSAHS performs a forward pass of the GNN to obtain pseudo-label predictions for all target nodes. The cost of an $L$-layer GNN forward pass consists of node-wise feature transformations and edge-wise message passing, resulting in a complexity of $O\big(Ln_t F^2 + L|E_t|F\big)$.

*Step 2: Homophily estimation within the agreement neighborhood.* Homophily ratios are computed using only edges whose endpoints both belong to the agreement set. Since agreement nodes constitute a fraction $\theta_t$ of all target nodes, the expected number of such edges scales as $\theta_t^2|E_t|$. Thus, this step has complexity $O\big(\theta_t^2|E_t|\big)$.

*Step 3: Reweighting of existing edges among agreement nodes.* For agreement nodes identified as low-homophily, PSAHS reweights their existing edges depending on whether the connected nodes share the same predicted class. Each edge incident to an agreement node is processed once, yielding a complexity of $O(\theta_t|E_t|)$.

*Step 4: Selection of new agreement same-class neighbors.* For each class, the agreement nodes are ranked by their homophily ratios to determine which nodes will be added as new neighbors for same-class nodes. Let $n_{t,k}$ denote the number of target nodes predicted as label $k$. Sorting $\theta_t n_{t,k}$ agreement nodes within class $k$ requires $O((\theta_t n_{t,k}) \log(\theta_t n_{t,k}))$ time. Summing over all classes results in a total cost of $O((\theta_t n_t) \log(\theta_t n_t))$.

*Step 5: Insertion of new intra-class edges.* Finally, PSAHS inserts new edges between agreement nodes that are likely to belong to the same class. The number of such edges is proportional to the fraction of heterophilous edges among agreement nodes, resulting in a complexity of $O(\theta_t|E_t|(1 - h_t))$, where $h_t$ denotes the average homophily of target nodes.

Combining all steps above, the per-iteration complexity of target-graph adjustment is

$$O\big(LF^2 n_t + LF|E_t| + (\theta_t n_t) \log(\theta_t n_t)\big).$$

Since $L$ and $F$ are small constants and $\theta_t < 1$ in all experiments, PSAHS exhibits favorable scalability and remains efficient even for large target graphs.

### A.5.3. SCALABILITY ANALYSIS

We further evaluate the scalability of PSAHS through controlled experiments on synthetic graphs with increasing sizes. The source-domain homophily is fixed at $0.832$, while the target-domain homophily is set to $0.3$ to simulate a challenging heterophilous target scenario. For each graph size, we report the following metrics: (i) average runtime per iteration, (ii) average memory consumption, (iii) graph sparsity before and after adjustment, and (iv) node classification accuracy. Table 4 summarizes the experimental results.

As shown in Table 4, classification accuracy improves substantially as the graph size increases from 300 to 3,000 nodes and then stabilizes at around $95\%$ for larger graphs. This indicates that PSAHS effectively leverages richer structural information while maintaining stable performance at scale.

Both runtime per iteration and memory consumption increase smoothly with graph size. The observed growth is approxi-

*Table 4.* Scalability results on synthetic graphs of increasing size.

| Graph Size | #Edges | Avg Time (s) | Avg Memory (MB) | Accuracy (%) | Sparsity (Original) | Sparsity (Adjusted) |
|---|---|---|---|---|---|---|
| 300 | 365 | 0.02 | 13.6 | $82.13 \pm 0.85$ | 8.14e-3 | 9.74e-3 |
| 1500 | 9030 | 0.03 | 60.4 | $90.10 \pm 1.20$ | 8.03e-3 | 1.10e-2 |
| 3000 | 36104 | 0.04 | 130.7 | $93.31 \pm 5.44$ | 8.03e-3 | 1.12e-2 |
| 6000 | 143905 | 0.05 | 294.9 | $95.25 \pm 2.68$ | 8.00e-3 | 1.14e-2 |
| 9000 | 323879 | 0.07 | 464.7 | $94.76 \pm 7.21$ | 8.00e-3 | 1.18e-2 |
| 12000 | 576063 | 0.12 | 823.1 | $96.93 \pm 3.23$ | 8.00e-3 | 1.18e-2 |

mately linear in the number of nodes and sublinear in the number of edges, which is fully consistent with the theoretical complexity analysis. Even for the largest graph with 12,000 nodes, the per-iteration runtime remains well below one second.

Although PSAHS introduces additional homophilous edges during structure adjustment, the adjusted graphs remain highly sparse. The sparsity after adjustment increases only marginally compared to the original graphs, and the total number of edges remains of the same order. This confirms that PSAHS avoids uncontrolled graph densification.

Overall, both theoretical analysis and empirical results demonstrate that PSAHS scales efficiently in terms of runtime and memory, preserves graph sparsity despite structural refinement, and achieves substantial accuracy improvements without compromising computational feasibility.

# B. Complementary Experiments

## B.1. Dataset

### B.1.1. SYNTHETIC DATASET

We present the data generation procedure in the source domain and target domain. We generate the node attributes with mixture Gaussian distribution consisting of three classes, where each class has the same number of nodes. In the source domain, the node attributes are drawn from class-specific 10-dimension Gaussian distributions: the means of the three Gaussians are $[-1, 0, 0, 0, 0, 0, 0, 0, 0, 0]$, $[1, 0, 0, 0, 0, 0, 0, 0, 0, 0]$ and $[0, 1, 0, 0, 0, 0, 0, 0, 0, 0]$ for the source domain, and $[-1.5, 0.5, 0, 0, 0, 0, 0, 0, 0, 0]$, $[1.5, -0.5, 0, 0, 0, 0, 0, 0, 0, 0]$ and $[0.5, 1.5, 0, 0, 0, 0, 0, 0, 0, 0]$ for the target domain.

The covariance matrices for the three Gaussians are different random rotations of three diagonal matrices:

$$\begin{bmatrix} 4 & 0 & 0 & 0 & 0 & 0 & 0 & 0 & 0 & 0 \\ 0 & 4 & 0 & 0 & 0 & 0 & 0 & 0 & 0 & 0 \\ 0 & 0 & 4 & 0 & 0 & 0 & 0 & 0 & 0 & 0 \\ 0 & 0 & 0 & 4 & 0 & 0 & 0 & 0 & 0 & 0 \\ 0 & 0 & 0 & 0 & 4 & 0 & 0 & 0 & 0 & 0 \\ 0 & 0 & 0 & 0 & 0 & \frac{1}{4} & 0 & 0 & 0 & 0 \\ 0 & 0 & 0 & 0 & 0 & 0 & \frac{1}{4} & 0 & 0 & 0 \\ 0 & 0 & 0 & 0 & 0 & 0 & 0 & \frac{1}{4} & 0 & 0 \\ 0 & 0 & 0 & 0 & 0 & 0 & 0 & 0 & \frac{1}{4} & 0 \\ 0 & 0 & 0 & 0 & 0 & 0 & 0 & 0 & 0 & \frac{1}{4} \end{bmatrix} \begin{bmatrix} 1 & 0 & 0 & 0 & 0 & 0 & 0 & 0 & 0 & 0 \\ 0 & \frac{7}{9} & 0 & 0 & 0 & 0 & 0 & 0 & 0 & 0 \\ 0 & 0 & \frac{5}{9} & 0 & 0 & 0 & 0 & 0 & 0 & 0 \\ 0 & 0 & 0 & \frac{1}{3} & 0 & 0 & 0 & 0 & 0 & 0 \\ 0 & 0 & 0 & 0 & \frac{1}{9} & 0 & 0 & 0 & 0 & 0 \\ 0 & 0 & 0 & 0 & 0 & \frac{1}{9} & 0 & 0 & 0 & 0 \\ 0 & 0 & 0 & 0 & 0 & 0 & \frac{1}{3} & 0 & 0 & 0 \\ 0 & 0 & 0 & 0 & 0 & 0 & 0 & \frac{5}{9} & 0 & 0 \\ 0 & 0 & 0 & 0 & 0 & 0 & 0 & 0 & \frac{7}{9} & 0 \\ 0 & 0 & 0 & 0 & 0 & 0 & 0 & 0 & 0 & 1 \end{bmatrix} \begin{bmatrix} 4 & 0 & 0 & 0 & 0 & 0 & 0 & 0 & 0 & 0 \\ 0 & \frac{1}{4} & 0 & 0 & 0 & 0 & 0 & 0 & 0 & 0 \\ 0 & 0 & 4 & 0 & 0 & 0 & 0 & 0 & 0 & 0 \\ 0 & 0 & 0 & \frac{1}{4} & 0 & 0 & 0 & 0 & 0 & 0 \\ 0 & 0 & 0 & 0 & 4 & 0 & 0 & 0 & 0 & 0 \\ 0 & 0 & 0 & 0 & 0 & \frac{1}{4} & 0 & 0 & 0 & 0 \\ 0 & 0 & 0 & 0 & 0 & 0 & 4 & 0 & 0 & 0 \\ 0 & 0 & 0 & 0 & 0 & 0 & 0 & \frac{1}{4} & 0 & 0 \\ 0 & 0 & 0 & 0 & 0 & 0 & 0 & 0 & 4 & 0 \\ 0 & 0 & 0 & 0 & 0 & 0 & 0 & 0 & 0 & \frac{1}{4} \end{bmatrix}$$

Figure 6(a) shows the attribute density contours of each class for source (dashed) and target (solid) domains. Differences in separation, overlap, and orientation across domains reflect both mean and covariance shifts in the class-conditional attribute distributions. To further illustrate how attribute distributions vary across domains, Figure 6(b) presents the overall attribute density contours for each domain.

To generate the homophily shift, we fix one domain's node homophily by setting the intra-class probability $p = 0.02$ and the inter-class probability $q = 0.002$, which yields a graph homophily of 0.832. For the other domain, we iteratively decrease graph homophily by randomly selecting two homophilous edges $(u, u')$ and $(v, v')$, where $Y_u = Y_{u'} \neq Y_v = Y_{v'}$, removing them, and then reconnecting the heterogeneous edges $(u, v)$ and $(u', v')$ to decrease the graph homophily. This procedure is repeated until the graph homophily reaches desired values ranging from 0.8 to 0.1.

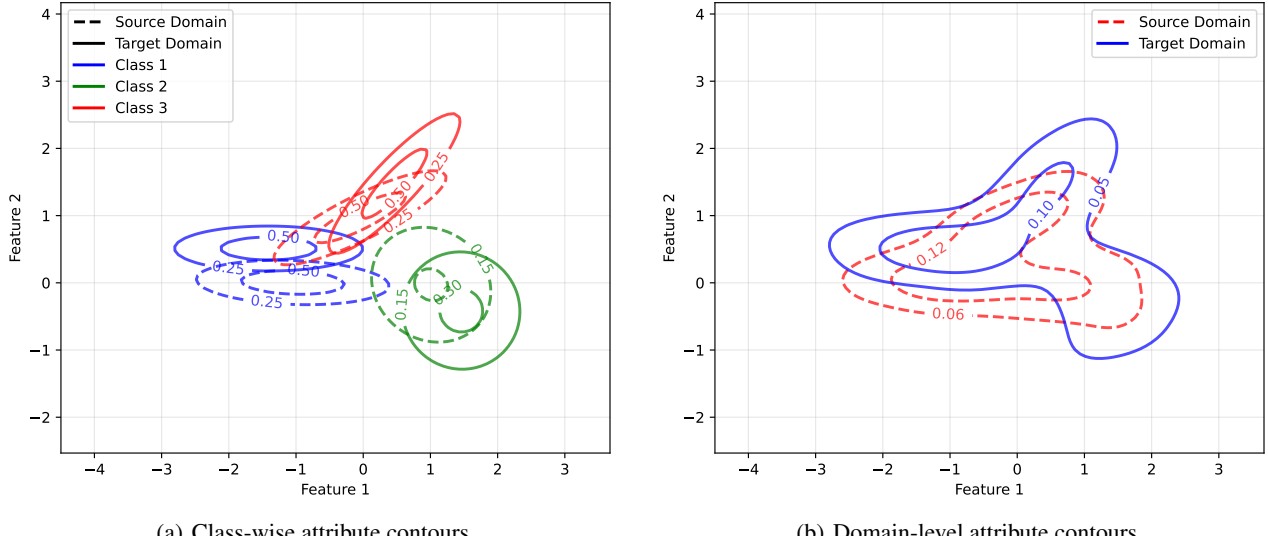

(a) Class-wise attribute contours                    (b) Domain-level attribute contours

*Figure 6.* Visualization of synthetic data.

*Table 5.* Dataset Statistics.

| Dataset | # Domains | # Nodes | # Edges | # Node_Homo | # Edge_Homo | # Feat Dims | # Labels |
|---------|-----------|---------|---------|-------------|-------------|-------------|----------|
| | USA | 1,190 | 27,198 | 0.3728 | 0.6978 | | |
| Airport | BRAZIL | 131 | 2,148 | 0.2478 | 0.4683 | 241 | 4 |
| | EUROPE | 399 | 11,990 | 0.2195 | 0.4048 | | |
| Blog | Blog1 | 2,300 | 66,942 | 0.3887 | 0.3991 | 8,189 | 6 |
| | Blog2 | 2,896 | 107,672 | 0.3728 | 0.4002 | | |
| Citation | DBLPv8 | 5,578 | 7,341 | 0.9750 | 0.9654 | 7,537 | 6 |
| | ACMv9 | 7,410 | 11,135 | 0.8179 | 0.8335 | | |
| | England | 7,126 | 35,324 | 0.5536 | 0.5560 | | |
| | Germany | 9,498 | 153,138 | 0.5974 | 0.6322 | | |
| Twitch | France | 6,566 | 65,955 | 0.5716 | 0.5595 | 3,170 | 2 |
| | Russia | 4,385 | 37,304 | 0.6300 | 0.6176 | | |
| | Spain | 4,648 | 59,382 | 0.6137 | 0.5800 | | |
| | Portugal | 1,912 | 31,299 | 0.5945 | 0.5708 | | |

### B.1.2. REAL-WORLD DATASETS

We briefly describe the real-world datasets used in Section 5. Table 5 summarizes the basic statistics of each dataset, such as the number of nodes, edges, feature dimensions, and classes. In addition, we report the average node-level and edge-level homophily to quantify the degree of homophily in each dataset.

DBLP and ACM. DBLP and ACM are two paper citation networks constructed from the DBLP and ACM digital libraries, respectively. We use the processed DBLPv8 and ACMv9 datasets from (Wu et al., 2022), where DBLPv8 contains papers published after 2010 and ACMv9 covers papers from 2000 to 2010. Nodes represent papers, and edges correspond to citation relationships between papers. Each node is associated with an attribute vector extracted from paper content, and labels indicate the research topic of the paper. Although the two networks share the same label space, they differ substantially in node homophily and structural patterns due to variations in citation practices and temporal coverage.

Twitch. The Twitch dataset (Liu et al., 2024a) consists of social networks collected from different regions on the Twitch streaming platform. Nodes represent Twitch users, and edges denote social connections between users. Node attributes capture user profile information and activity statistics, while node labels indicate whether a user is a streamer. Different regions exhibit different degrees of homophily and neighborhood mixing, leading to heterophilous patterns in certain domains. These variations naturally induce homophily shift across domains, and we evaluate cross-domain generalization by training on one region and testing on another.

`Airport.` The `Airport` dataset (Ribeiro et al., 2017) is an air transportation network where nodes correspond to airports and edges represent flight connections between them. Node attributes describe airport-level features such as traffic statistics, and node labels indicate airport categories. Due to hub-and-spoke structures and international traffic patterns, the network exhibits strong heterophily, with nodes frequently connecting to airports of different categories. When different countries or regions are treated as distinct domains, both the degree of heterophily and structural connectivity patterns vary, resulting in homophily shift across domains.

`Blog.` The `Blog` dataset (Li et al., 2015) is a social network of blog authors, where nodes represent bloggers and edges indicate interactions or hyperlinks between blogs. Node attributes are derived from textual or profile features, and labels correspond to blog categories or interests. Different subsets of the network are regarded as distinct domains, exhibiting variations in homophily levels and attribute distributions.

### B.2. Experimental Setup

The experiments are implemented using the `PyTorch` platform on a workstation equipped with an `Intel(R) Core(TM) i7-14700K CPU@3.40GHz` and a `NVIDIA GeForce RTX 4080 16GB GPU`.

In the modules of graph adjustment, the parameter grid for the prescribed homophily level $h$ is $\{0.5, 0.6, \ldots, 0.9, 1.0\}$. To improve the quality of pseudo labels, we pretrain an auxiliary MLP classifier with a 128-64-64 architecture on the source domain, which provides pseudo labels from a complementary perspective.

For the GNN encoder $\phi$, we adopt a $L$-layer graph convolutional network (GCN), where $L$ ranges from 2 to 5 and the hidden dimension is selected from $\{32, 64, 128, 512\}$.

In the module of cross-domain representation alignment, both the GNN classifier head and the domain discriminator are implemented as two-layer MLPs with hidden dimensions chosen from $\{16, 32, 64, 128\}$. The hyperparameter $\gamma_{\text{RA}}$ in the optimized objective function (8) follows the schedule: $\min\{2/(1 + e^{-10p}) - 1, 0.1\}$, where $p$ changes from 0 to 1 as the training epoch grows, as described in (Ganin et al., 2016). We select the learning rate in $\{0.0001, 0.001, 0.003, 0.01\}$.

The total number of training epochs is set to 300. At early training stages, pseudo-labels in the target domain are not yet stable across model updates; therefore, we introduce a warm-up phase to prevent premature structural adjustment. Specifically, we introduce a starting epoch $e$ and a structure adjustment frequency $t$, so that the model can gradually adapt to the evolving graph structure instead of being continuously updated from the beginning. The starting epoch $e$ determines when to begin adjusting graph structure on the target graph, while the reweighting frequency $t$ specifies how often the edge weights are updated. The search grids for $e$ and $t$ are $\{100, 150, 200, 250\}$ and $\{1, 5, 10, 15\}$. We repeatedly train and test our model for five times with the same partition of dataset and then report the average of accuracy.

For all datasets and methods, we follow a widely adopted GDA evaluation protocol. Models are trained solely on the source graph. On the target graph, 20% of node labels are reserved for validation and used only for model selection and hyperparameter tuning, while the remaining 80% are used for testing. Importantly, target validation labels are never involved in parameter optimization or model training, ensuring that no information leakage occurs and that the learning setting remains unsupervised. This protocol is standard in prior GDA works to ensure stable evaluation.

### B.3. Optimal Hyperparameter

*Table 6.* Hyperparameter settings for different datasets.

| Dataset | $h$ | $e$ | $t$ | GNN Hidden Dim | $L$ | MLP Hidden Dim | Learning Rate |
|---------|-----|-----|-----|----------------|-----|----------------|---------------|
| Airport | 0.7 | 100 | 10 | 128 | 2 | 64 | 0.001 |
| Blog | 1.0 | 100 | 10 | 128 | 4 | 128 | 0.001 |
| DBLP-ACM | 1.0 | 200 | 15 | 128 | 2 | 128 | 0.003 |
| Twitch | 0.7 | 100 | 15 | 128 | 3 | 128 | 0.001 |

### B.4. Analysis of Structural Adjustment.

We analyze the structural changes introduced by PSAHS to understand whether graph refinement leads to excessive perturbation. Empirically, the modification remains controlled and interpretable. On the target graph `Blog2`, PSAHS

adds 29,318 edges to the original 110,568 edges (including the self-loop edges), corresponding to a relative increase of approximately $26.5\%$, indicating moderate structural adjustment rather than aggressive graph rewiring. Moreover, the refinement is distributed across the graph instead of concentrating on a small subset of nodes: 1,388 out of 2,896 nodes ($47.9\%$) receive structural updates. For those modified nodes, the average number of added edges is 21.1, suggesting bounded local corrections. These observations indicate that PSAHS performs progressive and node-wise refinement, preserving graph sparsity and semantic consistency while enabling effective adaptation under homophily shift.

## B.5. Compared Methods

We compare our method, PSAHS, with the following representative approaches that address various aspects of graph-based domain adaptation:

- **UDA-GCN** (Wu et al., 2020): This method introduces a dual graph convolutional network that leverages both local and global consistency for more effective adaptation across domains. UDA-GCN focuses on improving feature propagation by integrating information from different parts of the graph.

- **ASN** (Zhang et al., 2021): ASN improves node representations by disentangling domain-specific and domain-invariant factors. It achieves this by using separate encoders for private and shared features, ensuring that the model learns more transferable representations between domains.

- **JHGDA** (Shi et al., 2023): JHGDA enhances domain adaptation by considering information from different hierarchical levels of the network. It uses a hierarchical pooling model to capture multi-level graph features, allowing for better alignment across domains.

- **StruRW** (Liu et al., 2023): StruRW addresses the challenge of neighborhood conditional shift by reweighting edges in the source graph. This edge reweighting helps to align neighborhood distributions between domains, improving the adaptation performance.

- **PairAlign** (Liu et al., 2024b): Similar to StruRW, PairAlign also reweights edges in the source graph to reduce the conditional shift in neighborhoods. However, it incorporates a pairwise alignment mechanism to better capture domain-specific and domain-invariant structures.

- **GraphAlign** (Huang et al., 2024): GraphAlign constructs a small yet transferable graph that aligns with the target domain through Maximum Mean Discrepancy (MMD) minimization. This approach preserves transferable knowledge by applying gradient matching techniques, helping to retain useful information for domain adaptation.

- **HGDA** (Fang et al., 2025b): HGDA mitigates the homophily shift by aligning multi-view feature representations across domains. It focuses on ensuring that the structure and features of the graph align across domains, preserving important homophilic relationships while adapting to new domains.

- **ADAlign** (Chen et al., 2026): ADAlign is a GDA method based on adaptive spectral distribution alignment. It introduces a Neural Spectral Discrepancy (NSD) defined on characteristic functions in the Fourier domain and employs a learnable frequency sampler to adaptively identify informative spectral components for cross-domain alignment. Through minimax optimization, ADAlign jointly aligns feature-structure discrepancies between source and target graphs.

## B.6. Optimal Prescribed Homophily Level $h$ under Noisy Pseudo-Labels

We analyze how the target-domain error bound in Theorem 4.1 depends on the prescribed homophily level $h$. For the source error term $\widehat{\mathcal{R}}_s^\gamma(g)$, since source labels are available, Theorem 4.2 implies that a larger $h$ (with $h = 1$ being maximal) minimizes the source-domain error. The homophily discrepancy term $W_1(h_s', h_t')$ admits a decomposition that separates the ideal supervised effect from the additional error introduced by pseudo-label noise. The representation discrepancy term $\text{IPM}(Z_s', Z_t')$ is largely insensitive to $h$ and is treated as approximately zero under near-perfect feature alignment.

Note that $h_t'(v)$ in $W_1(h_s', h_t')$ is computed after structure adjustment using pseudo-labels. Let $h_t'^*(v)$ denote the homophily

obtained if true target labels were available. Then

$$W_1(h'_s, h'_t) \leq \underbrace{\frac{1}{n_s n_t} \sum_{u \in V_s} \sum_{v \in V_t} \left| h'_s(u) - h'^*_t(v) \right|}_{=:W_1^I(h'_s, h'_t)} + \underbrace{\frac{1}{n_t} \sum_{v \in V_t} \left| h'^*_t(v) - h'_t(v) \right|}_{=:W_1^N(h'_s, h'_t)}. \tag{A.35}$$

Here, $W_1^I(h'_s, h'_t)$ denotes the ideal cross-domain homophily discrepancy under supervised target adjustment, which decreases with $h$, while $W_1^N(h'_s, h'_t)$ captures the noise-induced discrepancy due to pseudo-label errors, which increases with $h$. Note that $\mathcal{W}_1^N(h'_s, h'_t)$ depends only on target-side homophily estimation error induced by pseudo-labels; we keep the notation $\mathcal{W}_1^N(h'_s, h'_t)$ to emphasize its role as a component of the overall discrepancy $\mathcal{W}_1(h'_s, h'_t)$.

**Theorem B.1.** *The term $W_1^I(h'_s, h'_t)$ and $W_1^N(h'_s, h'_t)$ decrease and increase with $h$, respectively.*

*Proof of Theorem B.1.* For the term $W_1^I(h'_s, h'_t)$, we further decompose it into

$$W_1^I(h'_s, h'_t)$$

$$= \frac{1}{n_s n_t} \left( \sum_{\substack{h_s(u) \leq h \\ h_t(v) \leq h}} \left| h'_s(u) - h'^*_t(v) \right| + \sum_{\substack{h_s(u) > h \\ h_t(v) > h}} \left| h'_s(u) - h'^*_t(v) \right| + \sum_{\substack{h_s(u) \leq h \\ h_t(v) > h}} \left| h'_s(u) - h'^*_t(v) \right| + \sum_{\substack{h_s(u) > h \\ h_t(v) \leq h}} \left| h'_s(u) - h'^*_t(v) \right| \right)$$

$$= \frac{1}{n_s n_t} \sum_{\substack{h_s(u) > h \\ h_t(v) > h}} \left| h_s(u) - h_t(v) \right| + \frac{1}{n_t} \sum_{h_t(v) > h} \left( h_t(v) - h \right) + \frac{1}{n_s} \sum_{h_s(u) > h} \left( h_s(u) - h \right), \tag{A.36}$$

where the second inequality follows from

$$h'_s(u) = \begin{cases} h, & h_s(u) \leq h, \\ h_s(u), & h_s(u) > h, \end{cases} \quad \text{and} \quad h'^*_t(v) = \begin{cases} h, & h_t(v) \leq h, \\ h_t(v), & h_s(v) > h, \end{cases} \tag{A.37}$$

Obviously, from (A.36), the term $W_1^I(h'_s, h'_t)$ is decreasing with respect to $h$.

For term $W_1^N(h'_s, h'_t)$ in (A.35), the discrepancy between $h'^*_t(v)$ and $h'_t(v)$ arises from the difference between two structure-adjustment regimes. Specifically, $h'^*_t(v)$ corresponds to an oracle adjustment based on ground-truth labels, where all low-homophily nodes are refined, whereas $h'_t(v)$ corresponds to the practical pseudo-label-based adjustment, in which refinement is applied only to nodes identified as low-homophily under estimated labels.

To simplify the analysis, we make two assumptions. First, the homophily estimation error of a node $v$ originates solely from its own label prediction error, rather than from errors in its neighbors. Second, all nodes are assumed to be included in the agreement set, so that estimation errors are entirely attributed to incorrect pseudo-labels.

Under a false label prediction, i.e., $Y_v \neq \widehat{Y}_v$, the estimated homophily is inverted as

$$\widehat{h}_t(v) = 1 - h_t(v).$$

If $\widehat{h}_t(v) < h$, the adjustment enforces the estimated adjusted homophily to be $h$, which implies that the true homophily after adjustment becomes

$$h'_t(v) = 1 - h.$$

**Case 1:** $h \in [0.5, 1]$. For thresholds $h \geq 0.5$, the resulting true homophily after adjustment satisfies

$$h'_t(v) = \begin{cases} h, & h_t(v) \leq h, \ Y_v = \widehat{Y}_v, \\ 1 - h, & 1 - h \leq h_t(v) \leq h, \ Y_v \neq \widehat{Y}_v, \\ h_t(v), & h_t(v) < 1 - h, \ Y_v \neq \widehat{Y}_v, \\ h_t(v), & h_t(v) > h, \ Y_v = \widehat{Y}_v, \\ 1 - h, & h_t(v) > h, \ Y_v \neq \widehat{Y}_v. \end{cases} \tag{A.38}$$

Accordingly, term $W_1^N(h'_s, h'_t)$ in (A.35) can be decomposed as

$$W_1^N(h'_s, h'_t) = \frac{1}{n_t} \sum_{\substack{v:\, h_t(v) \geq h, \\ \widehat{Y}_v \neq Y_v}} \big( h_t(v) - 1 + h \big) + \frac{1}{n_t} \sum_{\substack{v:\, h_t(v) < 1-h, \\ \widehat{Y}_v \neq Y_v}} \big( h - h_t(v) \big) + \frac{1}{n_t} \sum_{\substack{v:\, 1-h \leq h_t(v) < h, \\ \widehat{Y}_v \neq Y_v}} \big( 2h - 1 \big). \quad \text{(A.39)}$$

where the above decomposition follows directly from (A.37) and (A.38).

**Case 2: $h \in [0, 0.5)$.** When $h < 0.5$, the homophily after adjustment under incorrect pseudo-labels becomes

$$h'_t(v) = \begin{cases} h, & h_t(v) \leq h,\ Y_v = \widehat{Y}_v, \\ 1 - h, & h \leq h_t(v) \leq 1-h,\ Y_v \neq \widehat{Y}_v, \\ h_t(v), & h_t(v) \leq h,\ Y_v \neq \widehat{Y}_v, \\ h_t(v), & h_t(v) > 1-h,\ Y_v = \widehat{Y}_v, \\ 1 - h, & h_t(v) > 1-h,\ Y_v \neq \widehat{Y}_v. \end{cases} \quad \text{(A.40)}$$

In this regime, term $W_1^N(h'_s, h'_t)$ reduces to

$$W_1^N(h'_s, h'_t) = \frac{1}{n_t} \sum_{\substack{v:\, h_t(v) > 1-h, \\ \widehat{Y}_v \neq Y_v}} \big( h_t(v) - 1 + h \big) + \frac{1}{n_t} \sum_{\substack{v:\, h_t(v) < h, \\ \widehat{Y}_v \neq Y_v}} \big( h - h_t(v) \big). \quad \text{(A.41)}$$

From (A.39) and (A.41), it is evident that term $W_1^N(h'_s, h'_t)$ increases monotonically with the homophily threshold $h$. Meanwhile, $W_1^N(h'_s, h'_t)$ decreases as pseudo-label accuracy improves. In the limiting case where all pseudo-labels are correct, i.e., $\{v \in V_t : Y_v \neq \widehat{Y}_v\} = \varnothing$, term $W_1^N(h'_s, h'_t)$ vanishes entirely, and the overall homophily discrepancy term $W_1(h'_s, h'_t)$ reduces to $W_1^I(h'_s, h'_t)$. $\qquad \square$

**Theorem B.2.** *If all target-domain nodes are assigned correct pseudo-labels, the target-domain error bound in* (10) *is minimized at the maximal prescribed homophily level $h = 1.0$. When pseudo-labels are imperfect, however, pseudo-label noise introduces a trade-off as $h$ varies, and the optimal homophily level may occur at some $h < 1.0$, particularly when the pseudo-label error rate is non-negligible.*

*Proof of Theorem B.2.* We analyze the behavior of the target-domain error bound in (10) as a function of the prescribed homophily level $h$.

When all pseudo-labels are correct, the noise-induced discrepancy term $W_1^N(h'_s, h'_t)$ vanishes. In this case, increasing $h$ monotonically reduces the empirical source margin loss $\widehat{\mathcal{R}}_s^\gamma(g)$ and the intrinsic homophily discrepancy term $W_1^I(h'_s, h'_t)$, as established in Theorems 4.2 and 4.3, respectively. Meanwhile, the representation discrepancy term $\mathrm{IPM}(Z'_s, Z'_t)$ is largely insensitive to $h$. Therefore, the target-domain error bound is minimized at the maximal homophily level $h = 1.0$.

When pseudo-label errors are present, the noise-induced term $W_1^N(h'_s, h'_t)$ becomes nonzero and typically increases with $h$, since stronger homophily enforcement amplifies the effect of erroneous labels. This introduces a trade-off between reducing intrinsic homophily mismatch and controlling noise amplification. As a result, the target-domain error bound may attain its minimum at an intermediate value $h < 1.0$. Such behavior is illustrated in Example B.3. $\qquad \square$

Theorem B.2 formalizes the role of pseudo-label accuracy in determining the optimal prescribed homophily level. In the absence of pseudo-label noise, enforcing maximal homophily ($h = 1$) is always optimal. With noisy pseudo-labels, however, the optimal $h$ becomes data-dependent and may remain below 1, gradually increasing toward 1 as pseudo-label accuracy improves. This analysis motivates setting a moderate lower bound (e.g., $h \geq 0.5$) for the parameter grid, with the final choice selected via a standard target-domain validation split (Liu et al., 2024b).

To further illustrate how pseudo-label errors affect the minimizer of the target-domain error bound with respect to the prescribed homophily level $h$, we consider a toy example in which the optimal $h$ can be computed explicitly under different pseudo-label accuracies.

**Example B.3.** Let the prescribed homophily level satisfy $h \in [0,1]$. Assume that the source- and target-domain node homophily ratios follow $h_s(u) \sim \mathrm{Beta}(1,2)$ and $h_t(v) \sim \mathrm{Beta}(2,1)$, respectively, and let $\rho$ denote the pseudo-label false rate. We further assume that the aggregated feature distribution is invariant across domains, so that term $\mathrm{IPM}(Z'_s, Z'_t)$ in the target-domain error bound vanishes.

Under this setting, the closed-form expressions of terms $\widehat{\mathcal{R}}_s^\gamma(g)$, $W_1^I(h'_s, h'_t)$, and $W_1^N(h'_s, h'_t)$, together with their monotonicity with respect to $h$, are summarized in Table 7. Specifically, terms $\widehat{\mathcal{R}}_s^\gamma(g)$ and $W_1^I(h'_s, h'_t)$ are decreasing functions of $h$, whereas $W_1^N(h'_s, h'_t) = \rho h^2$ on $h \in [0,1]$ is increasing in $h$, reflecting the growing penalty induced by pseudo-label errors when enforcing stronger homophily.

We next derive the closed-form expressions reported in Table 7.

*Table 7.* Closed-form expressions of terms $\widehat{\mathcal{R}}_s^\gamma(g)$, $W_1^I(h'_s, h'_t)$ and $W_1^N(h'_s, h'_t)$ and their monotonicity with respect to $h$.

| Term | Expression | Monotonicity in $h$ |
|---|---|---|
| $\widehat{\mathcal{R}}_s^\gamma(g)$ | $\frac{4}{3} - 2h^2 + \frac{2}{3}h^3$ | Decreasing |
| $W_1^I(h'_s, h'_t)$ | $\frac{7}{5} - \frac{10}{3}h + \frac{7}{3}h^2 - \frac{2}{3}h^4 + \frac{4}{15}h^5$ | Decreasing |
| $W_1^N(h'_s, h'_t)$ | $\rho\, h^2$ | Increasing |

By applying Theorem 4.2, we have

$$\mathcal{R}_s^\gamma(g) = \mathbb{E}_{h_s(u)}\big[2 - 2h'_s(u)\big] = 2 - 2\,\mathbb{E}_{h_s(u)}\big(\mathbf{1}\{h_s(u) \le h\}\, h + \mathbf{1}\{h_s(u) > h\}\, h_s(u)\big) = \tfrac{4}{3} - 2h^2 + \tfrac{2}{3}h^3. \quad \text{(A.42)}$$

Combining $W_1^I(h'_s, h'_t)$ in (A.36), $W_1^N(h'_s, h'_t)$ in (A.39), and the decomposition in (A.35), and letting $\rho$ be the proportion of false pseudo-labels in the target domain, we obtain the following decomposition:

$$
\begin{aligned}
W_1(h'_s, h'_t) &= \frac{1}{n_s n_t} \sum_u \sum_v \big|h'_s(u) - h'_t(v)\big| \\
&= \frac{1}{n_s n_t} \sum_{u:\, h_s(u) > h}\ \sum_{v:\, h_t(v) > h} \big|h_s(u) - h_t(v)\big| + \frac{1}{n_t} \sum_{v:\, h_t(v) > h} \big|h - h_t(v)\big| + \frac{1}{n_s} \sum_{u:\, h_s(u) > h} \big|h_s(u) - h\big| \\
&\quad + \mathbf{1}\{h \in [0.5, 1]\}\bigg(\frac{\rho}{n_t} \sum_{v:\, h_t(v) \ge h} \big(h_t(v) - 1 + h\big) + \frac{\rho}{n_t} \sum_{v:\, h_t(v) < 1-h} \big(h - h_t(v)\big) + \frac{\rho}{n_t} \sum_{v:\, 1-h \le h_t(v) < h} (2h - 1)\bigg) \\
&\quad + \mathbf{1}\{h \in [0, 0.5]\}\bigg(\frac{\rho}{n_t} \sum_{v:\, h_t(v) > 1-h} \big(h_t(v) - 1 + h\big) + \frac{\rho}{n_t} \sum_{v:\, h_t(v) < h} \big(h - h_t(v)\big)\bigg). \qquad \text{(A.43)}
\end{aligned}
$$

Replacing empirical summations with expectations with respect to $h_s(u)$ and $h_t(v)$ yields

$$
\begin{aligned}
\mathbb{E}\big[|h'_s(u) - h'_t(v)|\big] &= \mathbb{E}\big[|h_s(u) - h_t(v)|\, \mathbf{1}\{h_s(u) > h\}\, \mathbf{1}\{h_t(v) > h\}\big] \\
&\quad + \mathbb{E}\big[|h - h_t(v)|\, \mathbf{1}\{h_t(v) > h\}\big] + \mathbb{E}\big[|h_s(u) - h|\, \mathbf{1}\{h_s(u) > h\}\big] \\
&\quad + \mathbf{1}\{h \in [0.5, 1]\}\big(\rho\, \mathbb{E}\big[(h_t(v) - 1 + h)\, \mathbf{1}\{h_t(v) \ge h\}\big] + \rho\, \mathbb{E}\big[(h - h_t(v))\, \mathbf{1}\{h_t(v) < 1 - h\}\big] \\
&\qquad\qquad + \rho\, \mathbb{E}\big[(2h - 1)\, \mathbf{1}\{1 - h \le h_t(v) < h\}\big]\big) \\
&\quad + \mathbf{1}\{h \in [0, 0.5]\}\big(\rho\, \mathbb{E}\big[(h_t(v) - 1 + h)\, \mathbf{1}\{h_t(v) > 1 - h\}\big] + \rho\, \mathbb{E}\big[(h - h_t(v))\, \mathbf{1}\{h_t(v) < h\}\big]\big) \\
&= \tfrac{7}{5} - \tfrac{10}{3}h + \big(\tfrac{7}{3} + \rho\big)h^2 - \tfrac{2}{3}h^4 + \tfrac{4}{15}h^5. \qquad\qquad\qquad\qquad \text{(A.44)}
\end{aligned}
$$

Consequently, $W_1(h'_s, h'_t)$ can be written as a sum of a decreasing component $W_1^I(h'_s, h'_t)$ and an increasing pseudo-label-induced component $W_1^N(h'_s, h'_t) = \rho h^2$, as summarized in Table 7.

Fig. 7 plots the resulting target-domain error bound

$$\gamma \cdot \widehat{\mathcal{R}}_s^\gamma(g) + (1 - \gamma) \cdot \big(W_1^I(h'_s, h'_t) + W_1^N(h'_s, h'_t)\big) + \mathrm{IPM}(Z'_s, Z'_t),$$

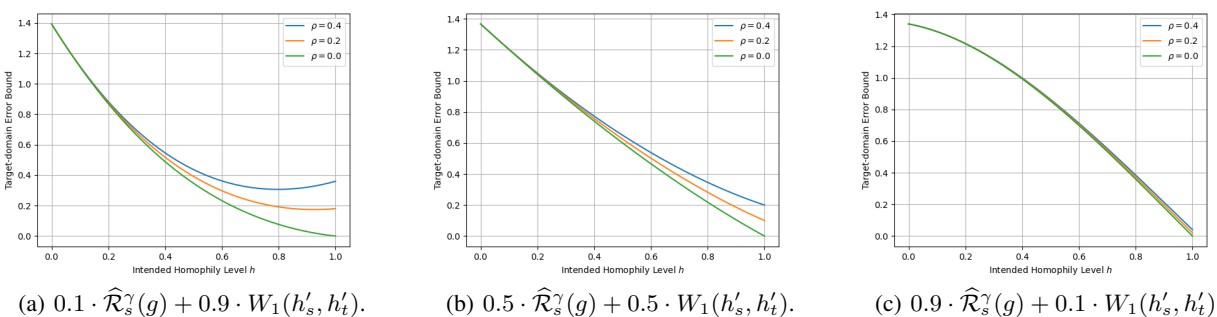

(a) $0.1 \cdot \widehat{\mathcal{R}}_s^\gamma(g) + 0.9 \cdot W_1(h_s', h_t')$.     (b) $0.5 \cdot \widehat{\mathcal{R}}_s^\gamma(g) + 0.5 \cdot W_1(h_s', h_t')$.     (c) $0.9 \cdot \widehat{\mathcal{R}}_s^\gamma(g) + 0.1 \cdot W_1(h_s', h_t')$.

*Figure 7.* Target-domain error bound as a function of the prescribed homophily level $h$ under different pseudo-label false rates $\rho$.

with trade-off parameter $\gamma \in (0,1)$ under different pseudo-label false rates $\rho$. When $\gamma = 0.1$, the minimizers are $h^* = 0.8, 0.9$, and $1.0$ for $\rho = 0.4, 0.2$, and $0.0$, respectively. In contrast, when $\gamma = 0.5$ or $\gamma = 0.9$, the minimizer consistently becomes $h^* = 1.0$ regardless of $\rho$.

Example B.3 demonstrates that the optimal prescribed homophily level $h$ is not necessarily $1.0$ in the presence of pseudo-label errors. When $\rho$ is large, the increasing penalty term $W_1^N(h_s', h_t') = \rho h^2$ counteracts the benefit of enforcing stronger homophily, shifting the minimizer smaller than but closer to $1.0$ (cf. Fig. 7(a)). As $\rho$ decreases, this penalty weakens and the minimizer moves back toward $h = 1.0$. Moreover, since the optimal $h$ across all trade-off settings consistently exceeds $0.5$, we restrict the search grid of $h$ in our experiments to $\{0.5, 0.6, \ldots, 1.0\}$.

