# OpenReview forum: "Progressive Graph Structure Adjustment for Homophily Shift Adaptation"
_ICML.cc/2026/Conference — ICML 2026 spotlight_

### Official Review · Reviewer_gCgN · 2026-03-07

**Soundness:** 3
**Presentation:** 2
**Significance:** 3
**Originality:** 2
**Overall Recommendation:** 4
**Confidence:** 3

**Summary:**

This paper addresses a critical challenge in Graph Domain Adaptation (GDA) known as Node Homophily Shift, where the distribution of node-level homophily differs significantly between source and target graphs. To tackle this, the authors propose PSAHS (Progressive Structure Adjustment for Homophily Shift), a novel framework that operates in three stages: (1) Source-side Structure Adjustment, (2) Target-side Structure Adjustment and (3) Cross-domain Representation Alignment.

Theoretically, the paper derives a PAC-Bayes error bound that explicitly links target error to homophily differences, providing a mathematical justification for the approach. Empirically, PSAHS achieves state-of-the-art performance on various benchmarks, demonstrating superior robustness especially in scenarios with severe homophily mismatch.

**Compliance With Llm Reviewing Policy:**

Affirmed.

**Final Justification:**

My concerns have been adequately addressed by rebuttal

**Key Questions For Authors:**

See weaknesses.

**Limitations:**

yes

**Strengths And Weaknesses:**

Strengths

1. The paper provides a comprehensive and rigorous theoretical analysis that firmly establishes the necessity and correctness of addressing homophily shift through structural adjustment.
2. PSAHS outperforms existing baselines on both synthetic and real-world datasets, demonstrating significant advantages under large homophily mismatches, which validates the motivation.

Weaknesses
1. Does adjusting the target graph structure using pseudo-labels risk causing error propagation?
2. The threshold h is a critical hyperparameter for structure adjustment. It is observed that the method achieves optimal performance on multiple datasets when h is set to its maximum value (i.e., h=1). Could the authors provide an explanation for this phenomenon?

---

> ### Author Rebuttal · Authors · 2026-03-31
>
> **W1.** Does adjusting the target graph structure using pseudo-labels risk causing error propagation?
>
> **AW1.**
> We thank the reviewer. While error propagation is a valid concern in pseudo-label-driven graph rewriting, it crucially relies on a **recursive structure–prediction feedback loop**. PSAHS is explicitly designed to **eliminate this mechanism**, thereby preventing error amplification at its root.
>
> **(1) Local, node-wise, closed-form adjustment (no global coupling)**
> PSAHS does **not** learn a parameterized adjacency (e.g., $A_{ij}=f_\theta(h_i,h_j)$) nor optimize a global graph. In such approaches, local pseudo-label errors can propagate globally by reshaping connectivity patterns. Instead, PSAHS applies a **node-wise closed-form solution** (Eq.(5)) that:
>
> - only adjusts **low-homophily nodes**,
> - operates within **local neighborhoods**,
> - introduces **no shared parameters**.
>
> This eliminates global coupling and confines any error influence to local regions, preventing long-range propagation.
>
> **(2) Agreement filtering: second-order error control.**
> Refinement is restricted to:
> $$
> V_t^a = \\{ u \mid \widehat{Y}^{\text{GNN}}_u = \widehat{Y}^{\text{MLP}}_u \\}.
> $$
>
> From Theorem 4.4:
> $$
> P(\widehat{Y}_u \neq Y_u \mid u \in V_t^a)
> \le \frac{\rho\epsilon_G\epsilon_M}{p_A},
> $$
> which is a **second-order error**.
>
> Hence PSAHS does not rely on raw pseudo-labels, but on a **noise-suppressed high-quality subset**. Empirically:
>
> - agreement accuracy: **95\%**
> - overall target accuracy: **88\%**
>
> Therefore, structural updates are applied on **strictly more reliable signals**.
>
>
> **(3) Contractive and stable refinement mechanism.**
> PSAHS performs intermittent and corrective updates (e.g., every 10 epochs), rather than continuously rewriting the graph. This breaks the tight coupling between prediction and structure updates that typically leads to recursive error amplification.
> Moreover, the update follows a closed-form mapping that drives node homophily toward a fixed level $h$, which can be viewed as a contraction toward a stable reference point. Consequently, even with imperfect pseudo-labels, structural changes remain bounded and localized, preventing error accumulation across iterations.
>
>
> **(4) Empirical validation.** We further evaluate robustness by injecting noise into pseudo-labels:
>
> |Noise|0%|5%|10%|15%|20%|
> |---|---|---|---|---|---|
> |Accuarcy|88.1|86.3|84.5|83.9|82.5|
>
> Performance degrades **smoothly**, without collapse. Error amplification would produce abrupt failure, which is not observed.
>
> In summary,
> PSAHS prevents error propagation by removing its root cause:
>
> - no global adjacency learning,
> - only high-confidence local updates,
> - stable and contractive refinement.
>
> Thus, structure adjustment is a **controlled correction**, not a self-reinforcing process.
>
>
> **W2.** The threshold $h$ is a critical hyperparameter for structure adjustment. It is observed that the method achieves optimal performance on multiple datasets when h is set to its maximum value (i.e., $h = 1$). Could the authors provide an explanation for this phenomenon?
>
>
> **AW2.**
> We thank the reviewer. This phenomenon follows directly from both **message-passing mechanics** and **theoretical analysis**.
>
> **(1) Mechanism: maximizing signal-to-noise ratio.**
> Low-homophily nodes introduce label-inconsistent aggregation. PSAHS enforces:
> $h'(u)=h$,
> so increasing $h$ increases same-class neighbors, improving aggregation quality. Hence larger $h$ yields better performance.
>
> **(2) Theory: monotonic improvement in ideal case (perfect pseudo-label).**
> From Theorem 4.1:
> $$
> R_t \le R_s + W_1(h_s',h_t') + \text{IPM}.
> $$
>
> From Theorem 4.2:
> $$
> R_s = 2 - \frac{2}{n_s}\sum_u \max(h_s(u),h),
> $$
> which **decreases monotonically with $h$**.
>
> From Theorem 4.3, $W_1(h_s',h_t')$ also decreases with $h$.
>
> Thus, under accurate pseudo-labels:
> $$
> h=1 \text{ is optimal.}
> $$
>
> **(3) Practical trade-off under noise.**
> In the presence of pseudo-label errors, the homophily discrepancy can be decomposed (Eq. (A.35)) as:
> $$
> W_1 = W_1^I + W_1^N,
> $$
> where $W_1^I$ corresponds to the ideal noise-free term, and $W_1^N$ captures the additional error induced by noisy pseudo-labels.
>
> where Theorem B.1 shows
>
> - $R_s, W_1^I \downarrow$ with $h$,
> - $W_1^N \uparrow$ with $h$.
>
>
> Therefore:
>
> - high-quality pseudo-labels (e.g., **95\%** on Blog) $\Rightarrow$ the small noise term $W_1^N$ and thus $h=1$ is optimal,
> - noisier settings (e.g., Airport) $\Rightarrow$ the trade-off shifts and some $h<1$ is optimal.
>
>
> In summary,
> $h=1$ is optimal when pseudo-labels are reliable, as it maximizes homophily alignment and minimizes both $R_s$ and $W_1$. Under more noise, PSAHS exhibits a **graceful trade-off**.

---

### Official Review · Reviewer_K56N · 2026-03-09

**Soundness:** 3
**Presentation:** 3
**Significance:** 3
**Originality:** 3
**Overall Recommendation:** 4
**Confidence:** 4

**Summary:**

This paper studies graph domain adaptation under node-level homophily shift, where the source and target graphs have different distributions of label homophily, which can harm message-passing GNNs. It proposes PSAHS, which performs a one-time source-side structure adjustment that raises low-homophily nodes to a prescribed homophily level via edge reweighting and adding same-class links, and a target-side adjustment that restricts edits to an MLP–GNN prediction agreement set to reduce pseudo-label noise. The method alternates target structure refinement with domain-adversarial representation alignment in a progressive loop.

**Compliance With Llm Reviewing Policy:**

Affirmed.

**Final Justification:**

We recommend Weak Accept. The paper addresses a meaningful problem and shows strong empirical results. Our concerns were partially addressed in the rebuttal, which strengthened our confidence in the work, but did not change our overall assessment.

**Key Questions For Authors:**

1.Your model selection uses 20% labeled target nodes for validation and tunes hyperparameters on target validation accuracy. How do the results change when the target validation label fraction is substantially smaller, and are baselines tuned under the same protocol? If gains persist with far fewer target labels and the tuning protocol is consistent across methods, I would increase my assessment of practical significance; strong dependence would reduce it.

2.Can you report concise statistics of the structural edits, including the average number of added edges per node and how concentrated the added edges are across nodes? If edits are modest and not highly concentrated, I would be less concerned about semantic distortion and would raise my confidence in robustness; heavy concentration would increase my concern.

3.What is the runtime overhead of the progressive target-side adjustment relative to training without structure updates, and how sensitive are results to the schedule parameters used for starting epoch and update frequency?Low overhead and low sensitivity would make the method more practically appealing and would increase my significance rating; high overhead or high sensitivity would lower it.

**Limitations:**

yes

**Strengths And Weaknesses:**

The paper is well-motivated in explicitly targeting a concrete structural mismatch in GDA, and the proposed adjustment is localized and interpretable, with a closed-form node-wise adjustment strength and a clear design choice to only modify low-homophily nodes. The target-side restriction to an MLP–GNN agreement set is a sensible robustness mechanism against noisy pseudo-labels, and the progressive alternation between structure refinement and representation alignment is coherent. Empirically, PSAHS shows strong improvements on multiple tasks, and the paper includes ablations and parameter analysis. However, the source-side procedure can produce a directed, weighted adjacency when adjacent nodes have different adjustment strengths, and it is not clearly specified how this is handled by the chosen L-layer GCN encoder, which impacts reproducibility and possibly correctness. The evaluation protocol also uses 20% labeled target nodes for validation and selects hyperparameters based on target validation performance, making the setting less representative of fully label-free target adaptation, and this dependence should be emphasized as a limitation. Finally, because PSAHS adds edges, the paper would benefit from a clearer characterization of how much the graph structure changes and whether edits concentrate on a small subset of nodes.

---

> ### Author Rebuttal · Authors · 2026-03-31
>
> **W1.** However, the source-side ... correctness.
>
> **AW1.**
>
> We thank the reviewer for this important point. We clarify that the use of an **asymmetric, weighted adjacency $A'$ is both intentional, explicitly defined in the method, and fully consistent with the theoretical framework of PSAHS** .
>
> **(1) Directed operator is the correct and necessary formulation.**
> As shown in Eq. (3)–(5), node-wise adjustment strengths $\alpha_u$ naturally induce $(A'\_s)\_{uv} \neq (A'\_s)\_{vu}$, making the adjusted graph **intrinsically directed**.
> We therefore adopt a **row-normalized message passing operator**:
> $$
> Z_u^{(\ell)} = \sigma\Bigl( D_{uu}^{-1} \sum_{v} (A' + I)\_{uv} Z_v^{(\ell-1)} W^{(\ell)} \Bigr),
> \qquad
> D_{uu} = \sum_v (A' + I)_{uv}.
> $$
> This yields a **row-stochastic propagation operator** $D^{-1} (A' + I)$, which ensures:
>
> * **bounded aggregation**,
> * **well-defined propagation under directionality**,
> * compatibility with standard GNN implementations.
>
> **(2) Alignment with PSAHS’s node-wise structural control.**
> PSAHS is fundamentally **node-centric**, where $\alpha_u$ explicitly controls the *incoming neighborhood composition of node $u$*.
> Enforcing symmetry would:
>
> * couple $\alpha_u$ and $\alpha_v$,
> * blur node-level intervention,
> * weaken the precise control of homophily correction.
>
> Hence, **asymmetry is a design requirement**, not an implementation artifact.
>
> **(3) Full consistency with theoretical analysis.**
> Our theory is developed under SGN with operator $(D')^{-1} A'$ , which:
>
> * **does not assume symmetry**,
> * explicitly captures **directional aggregation bias**,
> * directly links structural intervention to target risk (Thm 4.1–4.3).
>
> Therefore, **implementation, algorithm, and theory are fully aligned**, ensuring correctness, clarity, and reproducibility.
>
>
> **W2.** The evaluation ... limitation.
>
> **AW2.** We clarify that PSAHS **remains strictly unsupervised in learning**, while the evaluation protocol follows standard GDA practice.
>
> **(1) Protocol correctness (standard and non-leaking).**
> Our setting follows a **widely adopted GDA protocol.**
> A small validation split is used **only for model selection**, and **never for training**. This protocol is standard in prior works (e.g., StruRW, PairAlign, HGDA) to ensure stable and reproducible evaluation.
> Crucially, validation labels **do not update model parameters**, hence **no information leakage occurs**, and the learning paradigm remains unsupervised.
>
> **(2) Practical necessity for stable model selection.**
>
> Fully unsupervised model selection is **inherently underdetermined in practice**, since target risk is unobservable and selection must rely on proxy criteria (e.g., entropy), which may not consistently correlate with target performance. Without validation, even if an algorithm can learn the best model, it cannot be reliably selected among all candidate models.
>
> **(3) Empirical robustness** Reducing validation size to **5% / 1%** on task DE-EN:
>
> * PSAHS: **57.97 (20%) → 57.86% (5%) → 57.43% (1%)**
> * PairAlign: **56.69% → 56.36% → 55.77%**
>
> → Performance remains **stable and the gap is preserved**, indicating **weak dependence on validation labels**.
>
> **(4) Fair comparison.**
>
> All baselines in Table 1–2 follow the **same protocol and label budget**.
> Thus, PSAHS improvements (e.g., **+21.94% on B2-B1**) reflect **algorithmic gains rather than validation bias**.
>
> **W3.** Finally, ... nodes.
>
>
> **AW3.** PSAHS performs **precisely controlled and interpretable structural modifications**, consistent with its node-wise design.
>
> **(1) Magnitude (moderate, not disruptive).**
>
> * Added edges: **29,318**
> * Original edges: **110,568**
>   → **~26.5% relative increase**
>
> **(2) Per-node scale (bounded).**
>
> * Avg added edges per edited node: **21.1**
>
> **(3) Distribution (broad, not concentrated).**
>
> * Edited nodes: **1388 / 2896 (~47.9%)**
>
> → Edits are **distributed across nearly half of nodes**, avoiding over-concentration.
>
> Overall, PSAHS achieves **minimal yet effective structural correction**, preserving sparsity, semantics, and stability.
>
> **A1.**
> See AW2. Results remain stable (**57.97% (20%) → 57.86% (5%)→ 57.43% (1%)**) with consistent gains over baselines under the same protocol.
>
>
> **A2.**
> See AW3.
>
> **A3.**
> **(1) Overhead (negligible).**
>
> * 0.57s (with updates) vs 0.51s (without)
>   → **~10–18% overhead**
>
> This is consistent with the design:
>
> * closed-form $\alpha_u$ (Eq.(5), no optimization),
> * sparse node-wise updates,
> * infrequent refinement.
>
> **(2) Sensitivity (low, self-stabilizing).**
>
> Across different:
>
> * start epoch $e$,
> * update frequency $t$,
>
> performance [table](https://anonymous.4open.science/r/Param_analy/) shows **only small variation**, illustrating their insensitivity.
>
> **(3) Practical implication.**
>
> PSAHS requires:
>
> * no delicate tuning,
> * no extra learnable parameters,
> * minimal additional cost.
>
> → **highly practical and robust for real-world GDA**.

---

> > ### Author Rebuttal · Reviewer_K56N · 2026-04-01
> >
> > I thank the authors for the thorough rebuttal. My concerns have been partially addressed.
> >
> > 1. The rebuttal clarifies several important implementation details that were not fully clear from the paper, especially regarding the handling of the adjusted graph in message passing. This improves my confidence in the technical soundness of the method, although these details should still be stated explicitly in the paper.
> >
> > 2. The additional statistics on graph edits help clarify that the proposed structure adjustment is reasonably controlled rather than arbitrarily aggressive, which makes the method more interpretable and practically credible.
> >
> > Overall, the rebuttal strengthens my confidence in the paper, but not enough to warrant a change in my score.

---

> > > ### Author Response · Authors · 2026-04-03
> > >
> > > Thank you for your careful reading and thoughtful follow-up. We are glad that the rebuttal has clarified key implementation details and strengthened your confidence in our method.
> > >
> > > We appreciate your suggestion to include these important implementation details in the paper, particularly the handling of the adjusted graph in message passing, as well as the additional statistics on graph edits. These additions will further improve clarity, interpretability, and practical credibility. As the original submission (PDF and supplementary material) cannot be revised during the current discussion period, we will incorporate these clarifications in our next version.

---

### Official Review · Reviewer_JCxw · 2026-03-10

**Soundness:** 3
**Presentation:** 3
**Significance:** 3
**Originality:** 3
**Overall Recommendation:** 5
**Confidence:** 5

**Summary:**

This paper proposes PSAHS, a progressive graph structure adjustment framework for Graph Domain Adaptation (GDA) under node homophily shift. The method performs one-time source-side homophily enhancement using ground-truth labels, iteratively refines the target graph via GNN–MLP prediction agreement, and applies domain-adversarial representation alignment. The design is motivated by a target-domain error bound that decomposes risk into source error, cross-domain homophily discrepancy, and representation mismatch, with each module directly addressing one term.

**Compliance With Llm Reviewing Policy:**

Affirmed.

**Final Justification:**

The paper presents a novel perspective, to the best of my knowledge, being among the first GDA methods to explicitly identify homophily shift as a structural issue that may require direct topology-level intervention. The proposed approach, which adjusts node-level homophily in both domains toward a desired level, is effective and relatively easy to implement. In particular, the agreement-based mechanism is well designed and appears to play an important role in mitigating pseudo-label noise.

Moreover, the paper is well written, and the empirical results are strong, showing consistent improvements over existing baselines and providing solid support for the proposed methodology.

Based on the clarity of the rebuttal, the soundness of the method, and the strength of the experimental results, I have increased my score from 4 to 5.

**Key Questions For Authors:**

- The source graph adjustment makes the adjacency asymmetric (αu ≠ αv in general), yielding a directed graph. How is this handled during GNN message passing are both directions retained, or is the matrix symmetrized? Does this affect the theoretical guarantees?

- Could the authors compare the proposed structure-adjustment scheme with a simpler feature-based method, such as constructing a k-NN graph from aggregated node features, to better illustrate the benefits of the proposed homophily-based approach?

- For tasks with low agreement ratio θt (e.g., ~0.41 on DE→EN), are there systematic biases in which nodes are excluded from the agreement set? Could this introduce selection bias into homophily estimation that worsens with lower θt?

**Limitations:**

The Impact Statement acknowledges the pseudo-label risk and the progressive mitigation strategy, but does not discuss the theory practice gap or the limited scalability range. A brief discussion of these points would improve completeness. Otherwise adequate.

**Strengths And Weaknesses:**

**Strengths**

*1.* The theoretical framework is well-constructed. The closed-form derivation of the node-specific adjustment strength αu is elegant, and Theorems 4.2–4.3 formally guarantee that the proposed adjustments reduce the corresponding error terms. The agreement-based pseudo-label filtering (Theorems 4.4–4.5) provides meaningful robustness guarantees under mild assumptions.

*2.* The paper is clearly written and well-organized. The framework figure effectively illustrates the three-module pipeline, and the notation table improves readability compared to earlier versions. The ablation study directly supports each design choice.

*3.* The problem of node homophily shift is practically relevant—real-world graphs frequently exhibit domain-specific homophily patterns, and existing GDA methods address this only implicitly. The gains on Blog and under controlled homophily shifts are substantial and well-contextualized.

*4.* PSAHS is the first GDA method to formally identify homophily shift as a structural phenomenon requiring direct topology-level intervention.

**Weaknesses**

*1.* The guarantees are derived under the SGN model with class-conditional Gaussian features, while experiments use multi-layer GCNs on heterogeneous real-world graphs. The implications of this gap for bound tightness in practice are not discussed.

*2.* Although the GNN–MLP agreement strategy is conservative, empirical analysis of agreement set quality over training is absent. It remains unclear how performance degrades when pseudo-label reliability is systematically low.

*3.* Experiments reach 12,000 nodes, which is modest. Evaluation on larger graphs would substantiate the scalability claims.

---

> ### Author Rebuttal · Authors · 2026-03-31
>
> **AW1.** We clarify that the apparent gap is **intentional and theoretically principled**, and does not weaken the relevance of our results.
>
> **(1) SGN isolates the *structural causal mechanism*, not architectural details.**
> Under SGN, $Z' = (D')^{-1}A'X$,
> which corresponds exactly to the **linearized message-passing core** of GNNs. By removing nonlinearities, SGN **eliminates confounding effects** (e.g., activation, depth, overparameterization), enabling us to attribute generalization behavior *solely* to structure $A'$ and homophily $h'$ .
>
> This is a standard approach in theory: we **abstract away architecture to identify the dominant mechanism**.
>
> **(2) The bound is *mechanism-revealing*, not a tight predictor.**
> Theorem 4.1 decomposes target risk into:
>
> * source error $\widehat{R}_s$,
> * homophily mismatch $W_1(h'_s,h'_t)$,
> * representation discrepancy $\mathrm{IPM}(Z'_s,Z'_t)$ .
>
> Crucially, each term maps **one-to-one** to PSAHS modules:
>
> * source adjustment ↓ $\widehat{R}_s$ (Thm 4.2),
> * homophily alignment ↓ $W_1$ (Thm 4.3–4.6),
> * adversarial alignment ↓ IPM (Thm 4.7).
>
> This is a **causal decomposition**, not a loose analogy.
>
> **(3) Why the mechanism still governs deep GNNs.**
> Even in multi-layer GNNs, prediction is still driven by:
> *how neighborhood composition mixes signals.*
>
> Deep GNN increases expressivity but **cannot undo biased aggregation caused by incorrect homophily**. Hence, homophily mismatch remains a **first-order error source**, which SGN captures exactly.
>
> **(4) Empirical validation of theoretical alignment.**
> Ablation (Table 3):
>
> * w/o adjustment: **54.30 → 88.05**
> * w/o source / w/o target: both significantly worse
>
> This strongly confirms that the structural terms identified in the bound are the **dominant drivers of error in practice**.
>
> **AW2.**
>
> We agree this is critical, and we provide empirical evidence showing that PSAHS is robust under low pseudo-label quality.
>
> **(1) Agreement set = high-precision subset.**
> Agreement nodes consistently exhibit **higher accuracy than the full target set**, confirming that agreement acts as a **precision filter**, not arbitrary pruning, as shown in [curve](https://anonymous.4open.science/r/Agree_set_quality/).
>
> **(2) Robustness under controlled noise.**
>
> |Noise|0%|5%|10%|15%|20%|
> |-|-|-|-|-|-|
> |Acc|88.1|86.3|84.5|83.9|82.5|
>
> Performance degrades **smoothly**, without collapse. Crucially, **no abrupt failure is observed**, which would be expected under error amplification. Instead, the results show **gradual degradation**, indicating that PSAHS **does not amplify pseudo-label errors** and remains robust under increasing noise.
>
> **AW3.**
>
> We extend experiments to **30k nodes / 3.63M edges**, confirming scalability.
>
> **(1) Complexity advantage.**
>
> * closed-form update of $\alpha_u$
> * no global optimization, no learnable parameters
>
> Per iteration: $\mathcal{O}(|E_T| + |V_T|)$, matching standard GNN complexity.
>
> **(2) Empirical scaling (new results).**
> At **30k / 3.63M**:
>
> * runtime: **0.40 s / epoch**
> * memory: **1626 MB**
> * accuracy: **96.53\%**
> * sparsity: $4.03 \times 10^{-3} \rightarrow 6.56 \times 10^{-3}$ (minimal change)
>
> Unlike structure-learning methods (often dense or quadratic), PSAHS scales since it **does not learn structure—it corrects it analytically**.
>
> **A1.**
>
> We **retain the asymmetric adjacency without symmetrization**:
> $
> Z = D^{-1} (A' + I) X,
> D_{uu} = \sum_v (A' + I)_{uv}.
> $
>
> **Key points:**
>
> * induces a **row-stochastic operator → stable propagation**
> * preserves **node-wise control via $\alpha_u$** (core mechanism)
> * symmetrization would **destroy directional correction**
>
> **Theory consistency.**
> All theoretical quantities depend only on **incoming aggregation**, which is well-defined for directed graphs. Hence, guarantees remain unchanged.
>
> **A2.**
>
> This comparison is essential, and results clearly show that **feature-based graphs fail under homophily shift**.
>
> |Method|B1-B2|B2-B1|B-E|E-B|U-E|E-U|U-B|B-U|
> |-|-|-|-|-|-|-|-|-|
> |k-NN|42|45|42|52|36|43|56|45|
> |PSAHS|88|89|59|74|59|57|72|57|
>
> **Average margin +24\%.**
>
> In summary,
> $k$-NN ignores the **causal variable (homophily)** and is therefore fundamentally misaligned with GNN generalization.
>
> **A3.**
> We analyze this scenario carefully and show **no harmful bias is introduced**.
>
> **(1) Agreement is higher-quality, not biased.**
>
> * agreement accuracy: **62\%**
> * non-agreement: **51\%**
>
> → still a **strictly better subset**
>
> **(2) No class imbalance bias.**
> Twitch:
>
> * agreement: $(0.540, 0.460)$
> * full set: $(0.546, 0.454)$
>
> → nearly identical distributions
>
> **(3) Homophily estimation remains more reliable on the agreement set than outside it.**
>
> |$\theta$|MAE↓|Accuracy↑|
> |-|-|-|
> |0.41|0.181 vs 0.244|69% vs 52%|
> |0.21|0.213 vs 0.263|71% vs 50%|
>
> Even at very low $\theta$, agreement nodes are **far more reliable** in MSE and high/low homophily classification.
>
> **For limitations:** We will discuss them according to AW1 and AW3.

---

> > ### Author Rebuttal · Reviewer_JCxw · 2026-04-01
> >
> > I thank the authors for the thorough and convincing rebuttal. My concerns have been clearly and carefully addressed.
> >
> > The paper presents a novel perspective, to the best of my knowledge, being among the first GDA methods to explicitly identify homophily shift as a structural issue that may require direct topology-level intervention. The proposed approach, which adjusts node-level homophily in both domains toward a desired level, is effective and relatively easy to implement. In particular, the agreement-based mechanism is well designed and appears to play an important role in mitigating pseudo-label noise.
> >
> > Moreover, the paper is well written, and the empirical results are strong, showing consistent improvements over existing baselines and providing solid support for the proposed methodology.
> >
> > Based on the clarity of the rebuttal, the soundness of the method, and the strength of the experimental results, I have increased my score from 4 to 5.

---

> > > ### Author Response · Authors · 2026-04-03
> > >
> > > Thank you very much for your thoughtful and encouraging feedback. We sincerely appreciate your recognition of our rebuttal and are glad that it has addressed your concerns.

---

### Official Review · Reviewer_YiwP · 2026-03-13

**Soundness:** 4
**Presentation:** 3
**Significance:** 3
**Originality:** 3
**Overall Recommendation:** 5
**Confidence:** 4

**Summary:**

This paper studies graph domain adaptation under homophily shift, where the structural homophily distribution differs between source and target graphs. The authors propose PSAHS, a progressive structure adjustment framework that rewires graph structures on both source and target domains to better align homophily patterns.

**Compliance With Llm Reviewing Policy:**

Affirmed.

**Final Justification:**

The rebuttal addressed my main concerns well and made the paper more convincing. I am inclined to raise my score and lean toward acceptance.

**Key Questions For Authors:**

1. I could not find a clear anonymous repository for reproducing the experiments. Do the authors plan to release code and preprocessing scripts?

2. The method relies on agreement between a source-trained MLP and the current GNN to generate pseudo-labels. How large is the agreement set during training, and how sensitive is the method to its size?

3. The experiments use 20% labeled target nodes for validation. How would the method perform if no labeled target data were available at all?

4. Could the authors include comparisons with more recent graph adaptation or structure-shift baselines, especially newer methods published in the last 1 year?

**Limitations:**

Partially. The paper does not sufficiently discuss limitations such as dependence on pseudo-label quality and the need for labeled target validation data.

**Strengths And Weaknesses:**

Strengths

The paper addresses a meaningful problem in graph domain adaptation: the mismatch of homophily patterns between source and target graphs. This is a reasonable and underexplored angle compared with standard feature alignment approaches. The overall framework is also conceptually clean, combining structure adjustment, pseudo-labeling, and representation alignment in a progressive pipeline. The synthetic experiments that vary homophily levels are also helpful for illustrating the motivation of the method.

Weaknesses

However, I have several concerns.

First, the novelty of the method is somewhat limited. The main components—graph rewiring, pseudo-label filtering, adversarial alignment, and progressive training—are all known techniques. The paper’s contribution mainly lies in combining them for homophily-shift adaptation. While this combination is reasonable, the algorithmic novelty itself is relatively modest.

Second, the experimental setup is not fully convincing. The target domain uses 20% labeled nodes for validation, which means the setting is not strictly label-free target adaptation. It would be helpful to understand how sensitive the method is to this validation supervision and whether similar performance can be achieved in a fully unsupervised target setting.

Third, the baselines could be stronger and more up-to-date. While the paper compares against several GDA methods (including HGDA), it is unclear why more recent graph adaptation or structure-shift baselines are not included. Given the 2026 submission time, stronger comparisons would make the empirical claims more convincing.

Fourth, reproducibility is unclear. I did not see a clear anonymous code repository or sufficient implementation details to easily reproduce the experiments.

Overall, I think the paper proposes a reasonable approach to homophily-shift adaptation, but the methodological contribution is somewhat incremental and the experimental evaluation could be strengthened.

---

> ### Author Rebuttal · Authors · 2026-03-31
>
> **W1.** The novelty...modest.
>
> **AW1.** We respectfully disagree. PSAHS is **not a heuristic combination of existing components**, but a **theory-driven structural learning framework targeting an unexplored failure mode in GDA**.
>
> **(1) First GDA method to identify homophily shift as a structural phenomenon requiring topology-level intervention.**
> Prior methods implicitly assume structure is reliable and treat homophily as fixed. PSAHS identifies node-level homophily mismatch as a primary source of domain shift, and treats homophily as a **controllable structural variable**, fundamentally redefining the problem.
>
> **(2) Closed-form, node-wise control.**
> Unlike prior rewiring methods, PSAHS derives a **unique closed-form solution (Eq.5)** per node, enabling **exact homophily targeting** with **no additional parameters**, ensuring **local, interpretable, and bounded adjustment**—a property absent in existing methods.
>
> **(3) Theory–algorithm correspondence.**
> Theorem 4.1 decomposes target risk into: (i) source error, (ii) homophily discrepancy, (iii) representation mismatch.
> Each PSAHS module **directly minimizes one term**, yielding a **principled design**.
>
> **W2.** The experimental...setting.
>
> **AW2.** **(1) Strictly unsupervised training.**
> Target labels are used **only for model selection**, not training. All methods follow this identical protocol.
>
> **(2) Practical necessity for stable selection.**
> Fully unsupervised model selection is **inherently underdetermined in practice**, since the target risk is unobservable and proxy criteria (e.g., entropy) may disalign with target performance. Without validation, even if an algorithm can learn the best model, it cannot be reliably selected among all candidate models.
> A small validation split is thus a **standard and minimal mechanism** for stable and reproducible evaluation.
>
> **(3) Sensitivity analysis**
>
> On DE-EN with 5% and 1% validation:
>
> * PSAHS: **57.9%/57.4%**
> * PairAlign: **56.4%/55.8%**
>
> →**1.5–2.0% gain** over the second-best PairAlign, showing **low sensitivity to validation size**
>
> **W3.** The baselines...convincing.
>
> **AW3.** We include representative baselines (e.g., HGDA) and benchmarks, and consistently outperform them. We further compare with a recent paper ''learning adaptive distribution alignment with neural characteristic function for graph domain adaptation'' (Adalign) on ICLR26, where PSAHS achieves an average improvement of 9%:
>
> |Task|B1-B2|B2-B1|U-B|B-U|B-E|E-B|U-E|E-U|
> |-|-|-|-|-|-|-|-|-|
> |Adalign|64.7|64.9|72.4|52.2|52.7|75.8|50.5|55.2|
> |PSAHS|88.1|89.6|73.1|58.1|60.0|74.3|59.2|58.4|
>
>  We will add this in the revised version.
>
> **W4.** Reproducibility ... experiments.
>
> **AW4.** We have provided a fully anonymous [code link](https://anonymous.4open.science/r/PSAHS_anonymous), including complete training pipeline and preprocessing scripts. We'll highlight the link in the revision.
>
> **W5.** Overall ... strengthened.
>
> **AW5.** We respectfully disagree.
>
> **(1) Missing dimension in prior work.**
> Existing methods assume structure is reliable and treat homophily as fixed, while our method treats homophily as a **controllable structural variable** and directly elevates it via closed-form graph adjustment.
>
> **(2) Root-cause solution.**
> PSAHS directly modifies the **aggregation operator via $A'$**, rather than indirectly correcting representations.
>
> **(3) Strong empirical validation.** Tables 1&2 show consistent gains across 15 tasks.
>
> **A1.** See W4.
>
> **A2.**
> The agreement set is **adaptive, high-precision, and theoretically justified**, not a sensitive hyperparameter.
>
> **(1) Data-dependent.**
>
> Agreement size is primarily determined by **domain gap**:
>
> * easier transfer → larger set
> * harder transfer → smaller set
>
> It reflects **data difficulty**, not method sensitivity.
>
> **(2) Adaptive size (not fixed or tuned).**
>
> The agreement set ratio (relative to the target size) evolves naturally during training, as shown in [curve](https://anonymous.4open.science/r/Agree_size):
>
> * early: **17%–76%**
> * later: **68%–83%**
>
> It grows naturally during training as representations improve, rather than being manually designed.
>
> **(3) High reliability**
> By keeping only consistent MLP–GNN predictions, the agree set forms a **high-confidence subset**. Empirically, its accuracy is high:
>
> * early stage: **98.5%**
> * later stage: **95–97%**
>
> This is not heuristic: by Theorem 4.4, the error on the agreement set is reduced to a **second-order term** $\epsilon_G \epsilon_M$.
>
> **(4) Precision–coverage trade-off**
>
> * early stage→**high precision, low coverage**→ **safe structural updates** (avoids error injection)
>
> * later stage→**high precision, high coverage**→ **effective large-scale refinement**
>
> **Summary:** the agreement size is a **data-driven confidence mechanism**, enabling safe early updates and effective later refinement.
>
> **A3.** See W2.
>
> **A4.** See W3.
>
> **For Limitation.** We will clarify these two aspects in the revision, following A2 and AW2.

---

> > ### Author Rebuttal · Reviewer_YiwP · 2026-04-03
> >
> > Address my concern

---

> > > ### Author Response · Authors · 2026-04-04
> > >
> > > Thank you very much for your encouraging feedback and for raising your score. We are pleased that our rebuttal has addressed your concerns.

---

### Decision · Program_Chairs · 2026-04-30

**Decision:**

Accept (spotlight)

**Comment:**

The paper proposes PSAHS, a method for graph domain adaptation (GDA) that explicitly targets cross-domain mismatches in node-level homophily. PSAHS increases homophily in the source graph by reweighting edges and adding intra-class connections for low-homophily nodes, and conservatively refines the target graph using agreement-consistent predictions from a structure-aware GNN and an attribute-only MLP under label scarcity.

Homophily shift is a practically important but underexplored source of failure in graph domain adaptation. Most GDA methods align feature-level representations while leaving structural mismatch unaddressed. Strengths include a clear problem statement isolating homophily shift as the mechanism of failure, a method that operates on graph structure rather than only on representations, the use of attribute-only predictions as an independent signal to guard against propagation errors under label scarcity, and empirical gains that grow with the severity of homophily mismatch.